# Risk-taking unmasked: Using risky choice and temporal discounting to explain COVID-19 preventative behaviors

**Kaileigh A. Byrne**[1]*, **Stephanie G. Six**[1], **Reza Ghaiumy Anaraky**[2], **Maggie W. Harris**[1], **Emma L. Winterlind**[1]

**1** Department of Psychology, Clemson University, Clemson, South Carolina, United States of America,
**2** Department of Human-Centered Computing, Clemson University, Clemson, South Carolina, United States of America

* KaileiB@clemson.edu

**Data Availability Statement:** All data files are available from the Open Science Framework database (https://osf.io/xy6aj).

## Abstract

To reduce the spread of COVID-19 transmission, government agencies in the United States (US) recommended precautionary guidelines, including wearing masks and social distancing to encourage the prevention of the disease. However, compliance with these guidelines has been inconsistent. This correlational study examined whether individual differences in risky decision-making and motivational propensities predicted compliance with COVID-19 preventative behaviors in a sample of US adults ($N = 404$). Participants completed an online study from September through December 2020 that included a risky choice decision-making task, temporal discounting task, and measures of appropriate mask-wearing, social distancing, and perceived risk of engaging in public activities. Linear regression results indicated that greater temporal discounting and risky decision-making were associated with less appropriate mask-wearing behavior and social distancing. Additionally, demographic factors, including personal experience with COVID-19 and financial difficulties due to COVID-19, were also associated with differences in COVID-19 preventative behaviors. Path analysis results showed that risky decision-making behavior, temporal discounting, and risk perception collectively predicted 55% of the variance in appropriate mask-wearing behavior. Individual differences in general decision-making patterns are therefore highly predictive of who complies with COVID-19 prevention guidelines.

## Introduction

In the era of the COVID-19 pandemic, normal social activities like going to the mall or meeting with friends engender a certain level of risk. COVID-19, or coronavirus, is a contagious respiratory virus that spreads through close-contact airborne and droplet transmission [1]. To mitigate the spread of COVID-19, the Center for Disease Control and Prevention recommended that people wear face masks, avoid nonessential indoor activities, engage in social distancing by staying at least six feet apart from other people when in public places, and avoid in-person gatherings [2]. However, Americans have exhibited mixed responses to these guidelines

**Funding:** Clemson Universtiy Creative Inquiry Program # 1267. The funders had no role in the study design, data collection and analysis, decision to publish, or preparation of the manuscript. No authors received salary funding from this grant.

**Competing interests:** The authors have declared that no competing interests exist.

with some individuals strictly adhering to these recommendations while others choose not to heed them.

Indeed, a simple trip to the grocery store could become a highly polarizing event. For example, if a store required face masks to enter, it is not uncommon to see people wearing their masks incorrectly on their chin or removing them upon entry. Some people may silently feel outraged towards people that are not complying with CDC guidelines, while others may feel a sense of solidarity in defying this 'new norm'. The factors that predict whether or not individuals will engage in COVID-19 preventative behavior are not well understood. Some have proposed that compliance with mask-wearing and social distancing may be based less on scientific findings and more on political affiliation [3, 4]. Others have suggested that younger adults may feel invincible, thinking that they will not get sick from COVID-19 [5]; thus, another view is that age may be a factor that affects compliance. However, empirical research examining decision-making factors that influence compliance with mask-wearing and social distancing guidelines is lacking. Therefore, in addition to demographic factors, this study seeks to examine whether certain decision-making constructs, such as risky decision-making and temporal discounting, are predictive of compliance with appropriate mask-wearing and social distancing behaviors.

## Judgment and decision-making perspectives

This study examines the relationship between COVID-19 preventative behaviors and individual differences in four classic judgment and decision-making constructs: decision-making under risk, risk perception, the optimism bias, and temporal discounting. This pandemic is an emotion-fueled, unprecedented situation, and simple decisions to go to the store or socialize could have life and death consequences. Below we define each of the four constructs examined in this study and describe how they may be related to COVID-19 preventative behavior.

**Decision-making under risk.** A risk involves options in which the probabilities of each possible outcome are known, but the exact outcome itself remains unknown [6]. For example, a decision-maker might choose between an alternative that has a 100% probability of providing $5 or an alternative that has a 50% probability of providing $20 but also a 50% probability of yielding $0. The decision to comply with social distancing guidelines can be framed as a decision under risk. A decision-maker can choose the safe option of maintaining one's health by choosing to wear masks and social distance. Alternatively, a person could choose the risky option of interacting with others, in which he or she benefits from the enjoyment of social interactions at the possible cost of contracting COVID-19 and/or infecting others. While there is an element of uncertainty in knowing exactly *who* is infected in a given setting, each state's public health department makes its daily infection rates publicly available.

Decisions made under risk can be conceptualized based on expected values, or the probability of an outcome occurring multiplied by the potential value of that outcome. Sensitivity to expected value can provide objective information about decision-making behavior that maximizes the likelihood of reward [7]. Choosing options with higher expected values reflects increased sensitivity to differences in expected value between choice options [7, 8]. As evidenced by performance on decision-making paradigms such as the Iowa Gambling Task [9, 10] and the Cups Task [7, 8], this increased sensitivity can lead to reward maximization [7, 8]. People do not always base decisions solely on reward maximization however and instead, frequently choose options that maximize certainty over larger, uncertain gains (risk aversion) [11]. Psychological and economic studies have shown that risk preferences are often not monotonic, and risky decisions can change under different conditions [12–14]. Individual differences in risk aversion have been observed due to such factors as age, gender, and personality

[15–17]. Given that reward sensitivity and risk level can differentially influence decision-making, it is important to consider how both factors may affect COVID-19 preventative behavior. This study seeks to determine whether increased risky decision-making may be associated with decreased mask-wearing and social distancing behavior.

**Risk perception.**  In addition to actual risk decision-making behavior, perceived risk may also shape the way a person responds to COVID-19. People tend to perceive the risk of a phenomenon as high when it is outside of one's control, may have catastrophic potential or fatal outcomes, or when it is a new risk [18]. Risk perception is also influenced by the affect heuristic, in which people's positive or negative feelings towards an activity or phenomenon guides their evaluation of its risk [19]. When people believe that they can feel pleasure from engaging in an activity, such as engaging with friends despite being amid a pandemic, the perceived risk of that activity may be low. On the other hand, when individuals have a first-hand negative experience with a potentially risky event, such as watching someone pass away from COVID-19 complications, then they may feel more negatively towards that event and perceive the risk as high. When risk perception is high, people should want to see that risk minimized [20, 21]. However, it is unclear whether that also means that they are willing to take action to mitigate that risk themselves. In evaluating the perceived risk of COVID-19, there are clear opposing forces at play—the high benefit of social interactions may decrease the perceived risk of COVID-19, but the risk of a new, potentially fatal virus may increase the perceived risk. It is expected that individuals with first-hand COVID-19 experience will have higher risk perception and that higher risk perception will enhance COVID-19 preventative behavior.

**Optimism bias.**  The optimism bias reflects the belief that negative events have a lower likelihood of affecting oneself compared to other people, while positive events are more likely to affect oneself [22, 23]. This bias in risk perception occurs when people attempt to predict the likelihood of future events occurring and result in a disconnect between perceived and actual risk. The optimism bias was observed during risk evaluations for the H1N1 flu pandemic in 2009 [24, 25]. Similarly, it is reasonable to predict that, on average, individuals may perceive that their risk of contracting COVID-19 may be lower than the risk of one's peers contracting the virus.

**Temporal discounting.**  Temporal discounting, also known as delay discounting, refers to the tendency to prefer small immediate rewards over larger delayed rewards [26]. The subjective value placed on reward options tends to decline as the time delay for those rewards increases. For example, given the choice between receiving a small amount of money now (e.g., $5 now) or a larger amount of money after a time delay, such as getting $10 after one month, many people would prefer the former option [27]. From an economic perspective, such decisions are 'irrational' because the objective value of the delayed option is larger than the immediate option. However, from a psychological perspective, immediate rewards elicit more tangible positive emotions in the present compared to imagining how one *might* feel in the future [28, 29]. Representations of future rewards also tend to be more abstract, while immediate rewards are more concrete and vivid [30, 31]. Individual differences in temporal discounting have been shown to predict maladaptive health behaviors, including drug and alcohol use, unhealthy eating, general prophylactic health behaviors, and lack of exercise [32–35].

## Decision-making research for the H1N1 pandemic

The most recent pandemic recorded prior to COVID-19 was the H1N1 influenza outbreak in 2009. Previous research with H1N1 responses showed that people who perceived the risk of contracting H1N1 as high exhibited low risk-taking behaviors and high avoidance behaviors,

like avoiding heavily populated areas [36, 37]. Some studies showed that people who exhibited more signs of worry about contracting the virus tended to engage in more preventative measures [36, 38]. Another study indicated that people who resided in areas with a high concentration of the virus reported the belief in a higher likelihood of catching the virus but showed no signs of a higher degree of engagement of preventative behaviors [39]. In terms of demographics, risk-aversive behavior was associated with older age [37] and larger household size [39]. Additionally, previous research indicates that preventative behaviors decreased over time [37, 39], suggesting a decrease in risk perception and a subsequent increase in risky behaviors. While some research on preventative behaviors during the H1N1 pandemic exists, measures of risk-taking behaviors and delayed discounting and their relationship to H1N1 responses are far scarcer. This fundamental gap in research with prior pandemics motivated the design of the present study on COVID-19 preventative behaviors.

## Decision-making research for the COVID-19 pandemic

While research on the effects and perceptions of COVID-19 is in its very early stages, early work has begun to characterize COVID-19 risk perception, transmission-mitigation compliance behavior, and optimism biases. Recent research has demonstrated that some factors that can increase COVID-19 related risk perception include first-hand experience, prosocial values, trust in medical recommendations, individual knowledge about the virus, and political ideology [40]. Age has also been shown to influence COVID-19 risk perception such that older adults perceive the risk of contracting COVID-19 as lower than younger adults but exhibit heightened risk perception of dying from COVID-19 [41].

In addition to risk perception, emotional states and personality traits influence compliance with COVID-19 safety recommendations. Fear and anxiety surrounding COVID-19 are associated with increased hand-washing and social distancing recommendations [42]. In terms of personality, trait conscientiousness has been shown to increase the likelihood of compliance with COVID-19 prevention guidelines by over 30% [43], while antisocial traits are associated with diminished compliance [44]. Early work has also found evidence to support the presence of the optimism bias for contracting COVID-19 [45–47]. Collectively, early COVID-19 decision-making research suggests that pro-social behavior, personal experience, demographics, personality, and emotional factors shape individuals' perception of COVID-19 risk and transmission-mitigation compliance behavior. However, it remains unclear how individual differences in risk perception, risky decision-making, and temporal discounting influence compliance with COVID-19 preventative behaviors.

## Current study and hypotheses

As of December 2020, nearly 350,000 Americans have died from COVID-19, and over 20 million Americans have contracted the virus [1]. At the time this research was conducted, COVID-19 vaccines were not available to the general population, and one of the only ways for people to protect themselves from contracting COVID-19 was by engaging in mask-wearing and social distancing behaviors. Consequently, it is crucial to understand how specific decision-making tendencies can predict adherence to COVID-19 prevention guidelines.

This research seeks to elucidate how individual differences in risky decision-making, risk perception, the optimism bias, and temporal discounting can forecast compliance with COVID-19 prevention guidelines. These specific decision-making constructs may reflect the way that individuals evaluate the pandemic information they are exposed to and subsequently influence their decision to engage in COVID-19 preventative behavior or not. We predict that increased risky decision-making, decreased risk perception of COVID-19, greater temporal

discounting, and increased magnitude of the optimism bias will be associated with reduced compliance with COVID-19 preventative behavior.

## Materials and methods

An a priori power analysis was performed that included five demographic covariates (age, political affiliation, SES, negative financial experiences due to coronavirus, and experience with coronavirus) and four primary predictors (temporal discounting, advantageous gambles, disadvantageous gambles, and ambiguous gambles). The power analysis conducted in G*Power 3.1 indicated that to have 80% power to detect an effect at the $p = .05$ level with an effect size of $f^2 = .10$, a minimum of 172 participants would be needed. We anticipated an exclusion rate of 15% due to missed attention checks, incomplete responses, or duplicate responses. The goal was therefore to recruit at least 200 participants. This study was pre-registered through the Open Science Framework (OSF): https://osf.io/kavxr. The data are also available through OSF: https://osf.io/xy6aj

### Participants

This study was approved by the Institutional Review Board at Clemson University (IRB Approval Number 2020–220) before procedures were implemented. Participants were recruited using Amazon Mechanical Turk (MTurk), and data collection occurred in two waves: the first wave ($N = 220$) occurred from 9/7/2020–9/11/2020, and the second wave ($N = 200$) occurred from 12/29/2020–12/30/2020. Study participation was voluntary. A total of 420 participants were recruited to complete the study online. Previous work has shown that several experimental cognitive psychology paradigms conducted MTurk have a high degree of reliability with laboratory-based results [48]. The typical risky decision-making trends in which individuals tend to be more risk-seeking in loss contexts and risk-averse in gain contexts have also been observed in MTurk samples with minimal differences in effect sizes compared to studies performed in a laboratory [49]. Therefore, there is evidence that MTurk is a reliable way to collect data using standard decision-making paradigms.

Participants were compensated $3.50 for completing the study. To be eligible for study participation, participants needed to be between the ages of 18–90 and live in the United States. Participants were excluded from data analysis if they failed to pass one or more attention check questions ($n = 12$) or completed the study more than once ($n = 4$). Although participants were prevented from taking the study with the same MTurk Worker ID more than once, we identified several cases of duplicate IP addresses. Three attention check questions (e.g., "If you are reading this question, please choose Option C") were included throughout the study. These questions were modeled after instructional manipulation checks from previous research with unsupervised participants [49, 50]. Previous psychology research using MTurk and other online platforms have recommended using such attention check questions to improve the reliability of the data [49–51].

Thus, there were 404 participants (195 females; age range = 18–81, $M_{age} = 40.91$, $SD_{age} = 13.57$) in the final sample. Table 1 shows the participant characteristics of the sample.

### Design

This study entailed a correlational design in which we examined the relationship between temporal discounting and decisions on a risky choice task with several coronavirus-related behaviors and beliefs. Temporal discounting scores, the proportion of risky choices, and the difference in perceived risk when social distancing compared to not social distancing were used as predictors. Mask-wearing behavior, interpersonal social interactions, social distancing activities, optimism

**Table 1.**

| Demographic Characteristics | | |
|---|---|---|
| **Variable** | **Mean** | **Standard Deviation** |
| **Age** | 40.91 | 13.57 |
| **Years of Education** | 14.70 | 2.21 |
| **Gender** | **Number** | **Percentage** |
| Male | 208 | 51.19% |
| Female | 195 | 48.27% |
| Non-binary | 1 | 0.25% |
| **Race/Ethnicity** | | |
| Caucasian/White | 331 | 81.93% |
| African American/Black | 29 | 7.18% |
| Asian/Asian Indian | 22 | 5.44% |
| Hispanic/Latino | 17 | 4.21% |
| Other | 5 | 1.24% |
| **Political Affiliation** | | |
| Democrat | 186 | 46.04% |
| Republican | 125 | 30.94% |
| Independent/Other | 93 | 23.02% |
| **Region** | | |
| Southeast (SE) | 127 | 31.44% |
| Midwest (MW) | 84 | 20.79% |
| West | 81 | 20.05% |
| Northeast (NE) | 73 | 18.07% |
| Southwest (SW) | 39 | 9.65% |
| **Income Level** | | |
| <$30,000 | 175 | 43.32% |
| $30,000–$49,999 | 88 | 21.78% |
| $50,000–$99,999 | 118 | 29.21% |
| >$100,000 | 23 | 5.69% |
| **Personal Knowledge of Someone Who Contracted Coronavirus** | | |
| Yes | 214 | 52.97% |
| No | 190 | 47.03% |
| **Negative Financial Experiences due to Coronavirus** | | |
| Yes | 136 | 33.66% |
| No | 268 | 66.34% |

bias towards coronavirus, and perceived risk of engaging in public activities were included as outcome measures. Covariates in the study included age, political affiliation, geographic region, income level, negative financial experiences due to COVID-19, and personal health experience with COVID-19 (e.g., knowing someone who became ill or died from COVID-19, contracting COVID-19 themselves). Geographic region was a non-significant predictor in all analyses and was trimmed from all models for simplicity. Some deviations from the OSF pre-registration to the final study were made, such as the addition of the social distancing variables. Documentation of these differences are described in the S1 File for full transparency.

## Measures

**Demographics.** Participants provided information about their age, gender, political affiliation, income, and state of residence (Table 1). Income was coded into income level brackets:

<$30,000 (low), $30,000–$49,999 (lower middle), $50,000–$99,999 (middle), $100,00+ (upper middle/high) [52]. State of residence was coded into geographic regions: West, Southwest, Midwest, Southeast, Northeast. Participants were also asked questions regarding the negative effects of the pandemic on their work situation. Participants could use a checklist to indicate whether they had received a pay cut, been furloughed or lost their job, or were unable to find work because of COVID-19. Participants were queried about their personal experience with COVID-19 using Yes/No responses. Specifically, participants were asked whether they had tested positive for COVID-19, personally knew someone who had become symptomatic and/ or ill because of COVID-19, or personally knew someone who had passed away because of COVID-19.

**Mask-wearing behavior.** To assess appropriate mask-wearing behavior, participants were first asked to indicate the percentage of time across the past 4–8 weeks that they wore a mask in public settings (e.g., grocery stores, malls, restaurants) using a slider bar. Next, participants were asked to report the percentage of time they wore a mask above their mouth but below their nose. This question was used to indicate *incorrect* mask-wearing behavior. The incorrect mask-wearing behavior value was then subtracted from 100% to provide an index of the amount of time that participants correctly wore a mask in public. Next, we multiplied the percentage of time that participants wore a mask in public settings by the number of times participants wore a mask correctly to compute the percentage of time that participants engaged in appropriate mask-wearing behavior in public.

**Social distancing behavior.** Social distancing behavior was measured in two ways. First, interpersonal social interactions were quantified as the total number of people outside one's household with whom participants had physical, face-to-face interactions *without* wearing masks or social distancing in the past 14 days. Secondly, social distancing activities were quantified as the total number of times that participants engaged in the following activities in the past 30 days: (1) spent time in a group of more than 20 people, including activities such as church or social gatherings, (2) attended a small group hangout of 3 or more people, (3) ate at a dine-restaurant, (4) went to a mall or shopping center, (5) went to a hair salon, nail salon, or barbershop, and (6) worked out at a gym outside one's home. A composite sum score of these six activities was computed to form the social distancing activities measure. This metric is similar to the COVID States Project's Relative Social Distancing Index [53]. Higher scores indicate less social distancing. S1 Table in S1 File provides further information about participants' social distancing behavior.

**Perceived risk of activities in public settings.** Participants were asked to indicate how risky they believed a list of activities were at the present time, assuming that people were not social distancing. The list included seven activities: returning to in-person work, returning to in-person school, going for a walk in the park, going to a restaurant, traveling by plane, attending an indoor concert with 500+ people, and attending a college football game. Participants indicated their perceived risk of engaging in these activities on a 1 (*Not at All Risky*) to 5 (*Extremely Risky*) scale. Participants were also asked the same questions, but assuming that people ARE social distancing. However, results concerning these variables were largely similar for both questions. Thus, we report the results for responses assuming *no* social distancing as the dependent variable below for simplicity. The perceived risk of these activities was computed as the average reported risk score across the seven activities.

Secondly, we computed the difference in perceived risk when people are and are not social distancing. This measure provides information regarding how effective social distancing is and allows for examining risk compensation behavior. Higher values reflect greater perceived risk while not social distancing compared to when engaging in social distancing.

**Optimism bias.** The optimism bias refers to the cognitive bias that aversive events are less likely to affect oneself relative to one's peers [22, 23]. To examine optimism bias in relationship to COVID-19, we asked participants "What do you think is the likelihood that an average person your same age and gender will contract coronavirus in the next six months?". We then asked participants to indicate the likelihood that they themselves would contract coronavirus in the next six months. Participants responded using multiple-choice percentage options (Less than 10%, 10–20%, 20–30%, etc.). The optimism bias was operationalized as the likelihood of others contracting COVID-19 minus the likelihood that the participant would contract COVID-19. Higher values indicate that participants believed that other people would be more likely to contract COVID-19 than them.

**Temporal discounting.** A delayed discounting task was utilized to investigate participants' preferences for instant or delayed reward gratification. A significant body of research comparing the effects of real to hypothetical rewards has demonstrated that temporal discounting rates are highly similar under both conditions [54–57], which suggests that hypothetical rewards are a valid proxy for incentivized rewards in temporal discounting experiments. As such, participants were asked questions about non-incentivized, hypothetical situations regarding whether they would prefer a certain amount of money now or a larger sum after a certain number of days. Four time-delays (7 days, 30 days, 180 days, and 365 days) were presented in random order, and the starting amount was $5. Each question increased the previous monetary amount by $5 until reaching $30 for the instant reward choice. An example question was "Would you rather have $5 now or $30 after 7 days?". Participants' indifference points, or the smallest sum of money for which they first indicated their preference for the instant gratification reward over the delayed reward, was then recorded. To evaluate the overall preference for instant gratification against delayed gratification, an area under the curve (AUC) approach was employed. Smaller AUC values indicate a stronger preference for immediate gratification over larger delayed rewards.

**Risky choice task.** The risky choice task involved 36 non-incentivized, hypothetical gain-framed decisions and was similar to a descriptive gains-only version of the Cups Task [8]. Most recent studies have shown that decisions on risky choice tasks are not significantly altered under hypothetical compared to real reward conditions [58–61], though some exceptions have been observed [62]. In the task, participants were explicitly made aware of the probability of reward and reward magnitude. On each trial, two choices were presented: a sure option and a risky option. The 'sure' option involved a guaranteed amount of money. The 'risky' option involved a probability of a larger amount of money and a probability of receiving no money.

In line with previous research, the expected value—the product of the reward probability and magnitude—of the risky option was manipulated, which allowed for comparing interactions between expected value and risk on decision-making [8]. Therefore, this task distinguishes the advantageousness of choices, defined by selecting options with higher expected values, from general risk preference. Specifically, the task involved disadvantageous risky gambles ($n$ = 12 trials) in which the expected value for the risky gamble was lower than the sure option. An example of this type of gamble would be choosing between getting $100 guaranteed or an option offering a 50% chance of getting $150, but also a 50% chance of getting $0. Both advantageous risky gambles ($n$ = 12 trials) in which the expected value for the risky gamble was higher than the sure option and equal gambles ($n$ = 12 trials) in which the expected value for the risky and sure options was identical or nearly identical were also presented. Trials were pseudo-randomized once at the study outset. S2 Table in S1 File shows the full list of questions.

While the Cups Task involves 54 gain and loss trials of varying expected value levels (disadvantageous, advantageous, and equal), the present task used a modified gains-only task because the study predictions were localized to risk behavior and not loss aversion. The average proportion of risky gambles across all gambling types was computed for regression analyses and used as the primary analysis variable for this task. Additionally, following previous research using the Cups Task and similar risky choice paradigms [8, 63–70], the proportion of risky choices for each gamble type (risky advantageous, risky equal, and risky disadvantageous) was computed and used in follow-up analyses; this provides further information about whether sensitivity to expected values, as reflected by differential risk-taking in advantageous compared to disadvantageous decision contexts, influences the outcome variables.

**Supplemental COVID-19 questions.** Participants were asked about their beliefs regarding mask-wearing effectiveness of reducing the spread of coronavirus on a scale from 1 (*Not at all effective*) to 4 (*Very Effective*). Additional questions include 'Uncertainty brought on by COVID-19 coronavirus has caused me stress' and 'I am worried about getting COVID-19 coronavirus' using a 1 (*Strongly Disagree*) to 7 (*Strongly Agree)* scale. Table 2 shows descriptive information for these supplemental questions. Participants were also asked to indicate on a scale of 1 (*Not at all*) to 100 (*Extreme*) the extent to which they thought they were a risk-taker.

**Supplemental prosociality measures.** Emerging research between data collection periods suggested that prosociality is a predictor of compliance with physical distancing guidelines and mask wearing [40, 71–73]. In line with this research, two measures of pro-sociality were added in the second wave of data collection: (1) the Prosocial Behavioral Intentions Scale [74] and (2) a version of the Dictator Game. Further methodological description of task, and the results for this measure are described in the S1 File.

## Procedure

Participants were first presented with the online consent statement and were asked to indicate whether they did or did not voluntarily agree to participate in the study. Participants who specified that they did not consent were prevented from continuing with the study. Participants who indicated their online voluntary consent completed the questions in the following order: demographic items, several supplemental COVID-19 questions, optimism bias items, mask-

**Table 2.**

| Variable Descriptive Information | | | |
|---|---|---|---|
| **Dependent Variables** | *Mean* | *Standard Deviation* | *Range* |
| Appropriate Mask Wearing | 78.72 | 34.44 | 0–100 |
| Number of Interpersonal Social Interactions | 4.02 | 6.49 | 0–31 |
| Number of Non-Essential Activities | 16.54 | 36.12 | 0–186 |
| Optimism Bias | 7.90 | 15.73 | -60–90 |
| Perceived Risk (*not* social distancing) | 3.76 | 0.97 | 1–5 |
| Perceived Risk (*are* social distancing) | 3.05 | 0.89 | 1–5 |
| **Independent Variables** | | | |
| Temporal Discounting | 0.60 | 0.29 | 0.15–1.00 |
| Overall Proportion of Risky Choices | 0.20 | 0.19 | 0–0.92 |
| Proportion of Advantageous-EV Risky Choices | 0.32 | 0.26 | 0–1.00 |
| Proportion of Equal-EV Risky Choices | 0.16 | 0.21 | 0–1.00 |
| Proportion of Disadvantageous -EV Risky Choices | 0.12 | 0.19 | 0–0.83 |

*Note*. Perceived Risk refers to the perceived risk of engaging in public activities. EV refers to expected value.

wearing behavior questions, social distancing questions, and questions pertaining to the perceived risk of activities in public settings. Several unrelated filler questions were intermixed to avoid demand characteristics. Next, participants completed the temporal discounting task followed by the risky choice task. Participants ended the study by indicating the extent to which they believed they were a risk-taker using self-report.

## Data analysis

To characterize associations between the independent and dependent variables, bivariate correlations were first performed. Next, multiple linear regressions using ordinary least squares (OLS) were performed for each of the dependent variables (Appropriate Mask-Wearing Behavior, Social Distancing Behavior, Optimism Bias, and Perceived Risk). The between-subjects fixed-effect independent variables were Delay Discounting, Proportion of Risky Choices, and Perceived Risk. The data collection wave (September vs. December) was included as a fixed factor in the analyses. The covariates age, political affiliation, income level, negative financial experiences due to COVID-19, and personal health experience with COVID-19 were also included in the model. The following regression equation was used:

$$Y = \beta_0 + \beta_1 X_1 + \beta_2 X_2 + \beta_3 X_3 + \beta_4 X_4 + \beta_5 X_5 + \beta_6 X_6 + \beta_7 X_7 + \beta_8 X_8 + \beta_9 X_9 + \beta_{10} X_{10} + \epsilon$$

In the equation, $\beta_1 X_1 - \beta_3 X_3$ represent the predictors (Average Proportion of Risky Choices, Temporal Discounting, and Difference in Perceived Risk) and $\beta_4 X4 - \beta_{10} X_{10}$ are the covariates (Data Collection Wave, Age, Education, Income Level, Political Affiliation, Personal COVID-19 experience, and financial complications from COVID-19).

For the regression models, tests to determine whether the data met the assumption of collinearity indicated that multicollinearity was not a concern (VIFs range = 1.04–1.20). All descriptive, correlational, and regression analyses were performed using RStudio Version 1.2.5042 and SPSS version 26, and the exploratory path model was performed using MPlus. Standardized beta coefficients are reported for regressions; unstandardized betas are reported in the S1 File.

## Results

### Descriptives

Participants' appropriate mask-wearing behavior ranged from 0%–100% ($M = 78.72$, $SD = 34.44$). The average number of people that participants engaged with in the past 14 days without wearing a mask or social distancing was 4.02 ($SD = 6.49$), and the average number of times participants engaged in non-essential activities in the past 30 days was 16.54 ($SD = 36.12$). Table 2 shows additional descriptive information for the independent and dependent variables.

Additionally, independent samples t-tests showed that participants in the December data collection wave ($M = 83.03$, $SD = 31.61$) reported greater mask-wearing than those in the September data wave ($M = 76.61$, $SD = 36.53$), $t(402) = -2.48$, $p = .014$. December participants also reported engaging in fewer social interactions ($t(402) = 2.32$, $p = .021$) and non-essential public activities ($t(402) = 3.88$, $p < .001$) than the September participants, suggesting that compliance with COVID-19 preventative guidelines increased from September to December in this sample.

The perceived risk of engaging in such activities assuming that people were social distancing was significantly lower at 3.05 ($SD = 0.89$), $t(403) = 20.63$, $p < .001$. In terms of beliefs about COVID-19, 60.89% believed that COVID-19 reduced transmission risk for both oneself

**Table 3.**

| COVID-19 Belief Characteristics | | |
|---|---|---|
| *Demographics* | *Number* | *Percentage* |
| Participant tested positive for COVID-19 | 33 | 8.17% |
| Know someone who tested positive for COVID-19 | 224 | 55.45% |
| Know someone who was symptomatic due to COVID-19 | 203 | 50.25% |
| Know someone who died due to COVID-19 | 90 | 22.28% |
| *Beliefs* | *Number* | *Percentage* |
| Masks *only* reduce one's own risk of contracting COVID-19 from others | 41 | 10.15% |
| Masks *only* reduce other people's risk of contracting COVID-19 | 80 | 19.80% |
| Masks do not reduce the risk for anyone from contracting COVID-19 | 37 | 9.16% |
| Masks reduce the risk for others and oneself of contracting COVID-19 | 246 | 60.89% |
| Masks are *very* effective in reducing spread of COVID-19 | 200 | 49.50% |
| Masks are *moderately* effective in reducing spread of COVID-19 | 140 | 34.65% |
| Masks are *a little* effective in reducing spread of COVID-19 | 34 | 8.42% |
| Masks are *not at all* effective in reducing spread of COVID-19 | 30 | 7.43% |

and others (the remaining participants believed masks reduced the risk for either oneself only, others only, or no one) and 49.50% believed that masks were very effective in reducing the spread of COVID-19 (Table 3). The majority of participants expressed positive beliefs toward mask-wearing in preventing or slowing COVID-19 transmission.

In terms of risky decision-making, 17.08% of participants chose the safe option on all trials. By gambling context, 19.55% of participants chose the safe option in all advantageous expected value contexts, 46.78% chose the safe option in all equal expected value contexts, 57.43% chose the safe option in all disadvantageous contexts. In contrast, no participants chose the risky option on all trials.

To examine the optimism bias, a paired samples t-test was conducted to compare participants' estimation of the likelihood that their peers would contract COVID-19 compared to them. Results showed evidence of an optimism bias for contracting COVID-19, $t(403) = 10.09$, $p < .001$, $d = .31$. Participants believed that the likelihood of others contracting COVID-19 ($M = 40.69$, $SD = 25.04$) was 7.9% higher than the likelihood of contracting the virus themselves ($M = 32.80$, $SD = 25.34$).

## Correlations

Results revealed significant correlations between the proportion of all risky gambles and appropriate mask-wearing behavior, interpersonal social activities, non-essential social activities, perceived risk of public activities, and the optimism bias ($ps < .05$; Table 4).

Additionally, significant correlations between temporal discounting scores ($ps < .05$) and difference in perceived risk ($ps < .05$) with appropriate mask-wearing, interpersonal social interaction, non-essential social activities, and perceived risk were also observed. These relationships suggest that the lack of appropriate mask-wearing and social distancing is associated with increased risk-taking behavior and preference for immediate small rewards over larger, delayed rewards. Greater perceived risk under non-social distancing conditions was associated with more mask-wearing and social distancing behavior. Moreover, greater perceived risk of engaging in public activities and greater optimism bias are associated with decreased risk-taking behavior.

**Table 4.**

| *Correlational Analyses* | | | | | |
|---|---|---|---|---|---|
| | **Appropriate Mask Wearing** | **Interpersonal Social Interactions** | **Social Activities** | **Optimism Bias** | **Perceived Risk** |
| Delay Discounting Score | 0.35** | -0.31** | -0.41** | 0.03 | 0.14* |
| Perceived Risk Difference | 0.40** | -0.30** | -0.36** | 0.13* | 0.46** |
| Proportion of All Risky Gambles | -0.30** | 0.28** | 0.34** | -0.19** | -0.18** |
| Proportion of Risky Advantageous Gambles | -0.04 | 0.07 | 0.06 | -0.13* | -0.11* |
| Proportion of Risky Disadvantageous Gambles | -0.47** | 0.39** | 0.51** | -0.18** | -0.19** |
| Proportion of Equal Gambles | -0.35** | 0.32** | 0.39** | -0.21** | -0.18** |

**indicates significance at the $p < .001$ level

*indicates significance at the $p < .05$ level

Additional correlations were performed between perceived risk and actual COVID-19 preventative behaviors. Results indicated that the higher perceived risk of engaging in public activities, assuming *no* social distancing, was significantly correlated with appropriate mask-wearing behavior ($r = .414$, $p < .001$), number of interpersonal social interactions ($r = -.262$, $p < .001$), and weakly with engaging in non-essential social activities ($r = -.131$, $p < .01$). Perceived risk assuming that people *are* social distancing was also positively correlated with mask-wearing ($r = .142$, $p = .004$), but the magnitude of the relationship was much weaker than for perceived risk assuming no social distancing.

Further correlations were conducted between the optimism bias and COVID-19 compliance behaviors. Results showed that the magnitude of the optimism bias was associated with reduced social interactions ($r = -.205$, $p < .001$), participation in non-essential social activities ($r = -.201$, $p < .001$), and increased mask-wearing ($r = .210$, $p < .001$).

### Regression analyses

**Appropriate mask-wearing behavior.** The linear regression for Appropriate Mask Wearing Behavior revealed a significant relationship between Proportion of Risky Choices and Appropriate Mask Wearing ($\beta = -.199$, $p < .0001$) such that those who made more risky choices reported wearing masks less (Fig 1). Separate follow-up analyses within gambling context showed that those who made more risky choices in disadvantageous ($p < .0001$) and equal ($p < .0001$) gambling tasks tended to disregard appropriate mask-wearing behavior. There was no significant relationship between advantageous risky choice and mask-wearing behavior ($p = .575$). Furthermore, temporal discounting predicted mask-wearing such that those who prefer delayed over immediate rewards tended to engage in more appropriate mask-wearing behavior ($\beta = .215$, $p < .0001$). Additionally, perceived risk difference was also associated with greater mask-wearing behavior ($\beta = .280$, $p < .0001$).

In terms of demographic covariates, the results showed that individuals who had not experienced financial problems due to COVID-19 ($M = 87.61$, $SD = 26.95$) reported practicing appropriate mask-wearing behavior more than those who had faced financial problems ($M = 61.19$, $SD = 40.47$, $\beta = -0.231$, $p < .0001$). In addition, those without previous COVID experiences ($M = 83.93$, $SD = .29.44$) were more likely to practice appropriate mask-wearing than those with personal COVID-19 experience ($M = 74.08$, $SD = 37.80$, $\beta = -.100$, $p = .018$). Age ($p = .191$), Education ($p = .112$), Political Affiliation ($p = .623$), and Income Level ($p = .418$) were not significant predictors of mask-wearing behavior. The model results are reported

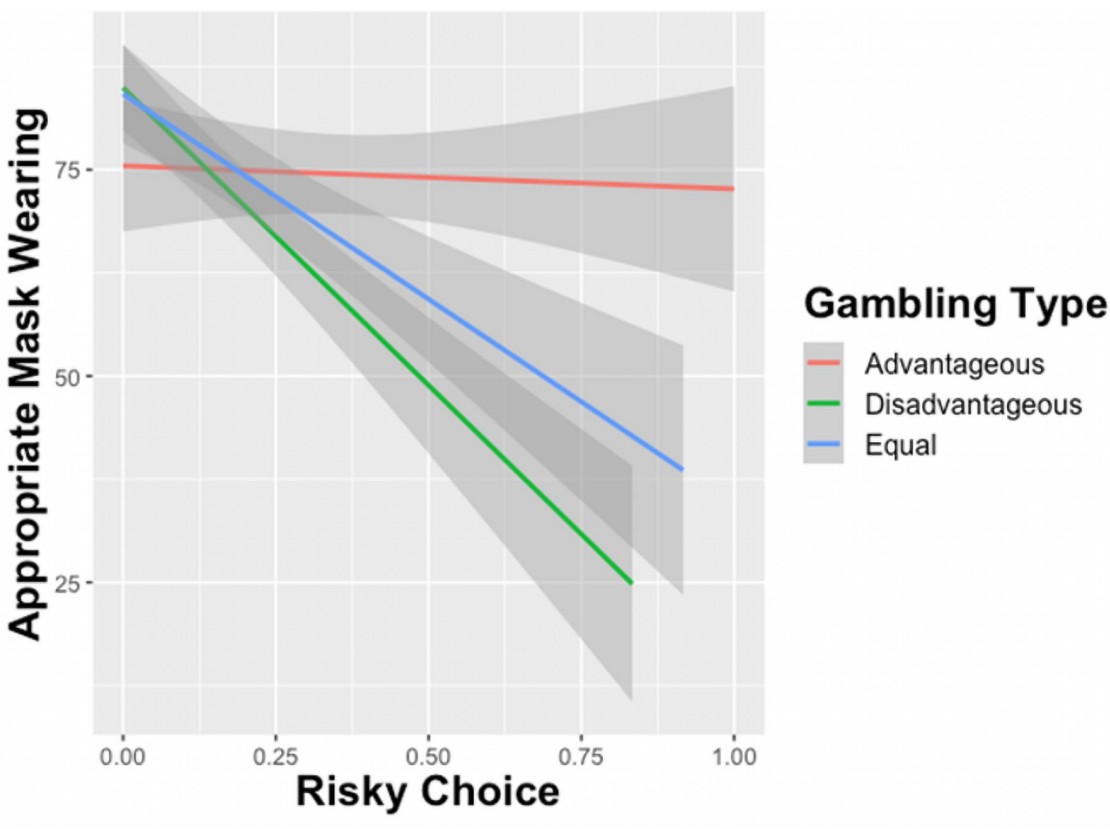

**Fig 1. Effect of risky choice on appropriate mask wearing behavior by gambling type.** Advantageous gambles indicate that the expected value for the risky choice was higher than the safe gamble. Disadvantageous gambles indicate that the expected value for the risky option was lower than the safe option, and the expected values for both safe and risky choices were the same for Equal gambles. Results show that choosing a higher proportion of Disadvantageous and Equal Gambles predicted less mask-wearing behavior.

in S3 Table in S1 File. Overall, 35.6% of the variance in appropriate mask-wearing behavior was explained by this model.

**Social distancing behavior.** Two linear regressions were conducted to examine the effect of temporal discounting and risky choice on social distancing behavior: one for Interpersonal Social Interactions and one for Non-Essential Social Activities. Participants that reported higher perceived risk when *not* social distancing engaged in fewer social interactions ($\beta$ = -.193, $p$ < .0001). In contrast, those who showed a higher preference for immediate rather than delayed rewards engaged in more interpersonal social interactions ($\beta$ = -.221, $p$ < .0001). In terms of risky decision-making, those who made more risky choices ($\beta$ = 185, $p$ < .001) tended to engage in more face-to-face social interactions without a mask. Separate follow-up analyses by gambling context showed that the effects of Risky Choice on Interpersonal Social Interactions were significant in the disadvantageous ($p$ < .001) and equal ($p$ < .001) contexts but not the advantageous ($p$ = .216) context. Fig 2 shows these effects.

Additionally, interpersonal social interactions were also significantly associated with Age, education level, COVID-19 experience, and whether or not COVID-19 had financially impacted the participants Surprisingly, level of education ($\beta$ = .160, $p$ = .0001), Age ($\beta$ = .096, $p$ = .035), personal experiences with COVID-19 ($\beta$ = .170, $p$ < .0001), and negative financial complications due to COVID-19 ($\beta$ = .135, $p$ = .003) were all associated with increased mask-

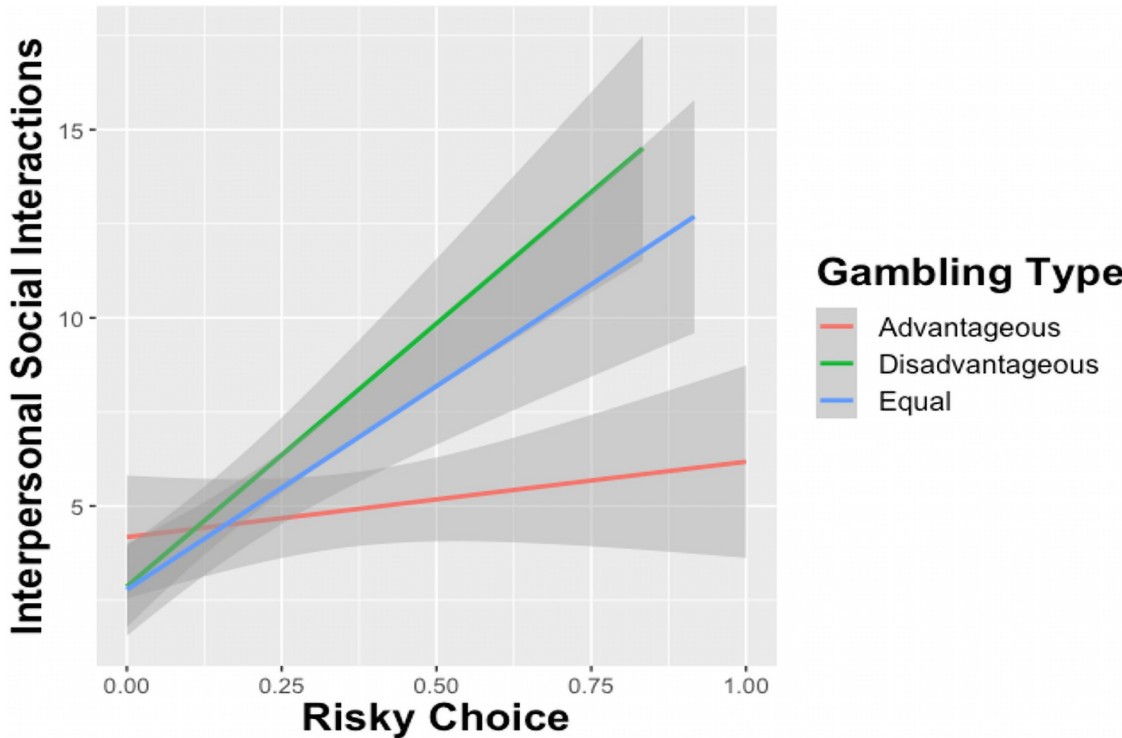

**Fig 2. Effect of risky choice on number of interpersonal social interactions by gambling type.** Advantageous gambles indicate that the expected value for the risky choice was higher than the safe gamble. Disadvantageous gambles indicate that the expected value for the risky option was lower than the safe option, and the expected values for both safe and risky choices were the same for Equal gambles. The results demonstrated that choosing a higher proportion of Disadvantageous and Equal Gambles predicted a greater number of interpersonal social interactions, meaning that participants engaged in less rigorous social distancing.

less social interactions and, thus, less social distancing. Regression results are reported in S4 Table in S1 File. The $R^2$ value for the overall model was .284.

The second linear regression examined predictors of Non-Essential Social Activities, like going to the mall or dine-restaurants. The results for this social distancing variable were largely consistent with results for Interpersonal Social Interactions. While difference in perceived risk were associated with less non-essential activities ($\beta$ = -.218, $p$ < .0001), both greater temporal discounting ($\beta$ = -0.282, $p$ < .0001) and increased proportion of risky choice ($\beta$ = .191, $p$ < .0001) on the gambling task predicted greater engagement in non-essential social activities, indicating less social distancing (Fig 3).

In terms of demographics, non-essential social activities were significantly associated with level of education, experience with COVID-19, and the financial impact of COVID-19. As level of education ($\beta$ = 0.189, $p$ < .0001), personal experiences with COVID-19 ($\beta$ = 0.120, $p$ = .002), and negative financial impact from COVID-19 ($\beta$ = 0.235, $p$ < .0001) increased, so did the number of non-essential social activities in which participants partook. Unlike Interpersonal Interactions, however, significant differences in Data Collection Wave were observed for non-essential social activities ($\beta$ = -0.106, $p$ = .009). S5 Table in S1 File shows the regression results. This model explained 43.8% of the variance in participation in non-essential social activities.

**Optimism bias.** The regression results (S6 Table in S1 File) showed that the proportion of risky choices predicted a reduced optimism bias ($\beta$ = -.158, $p$ = .002), but neither temporal

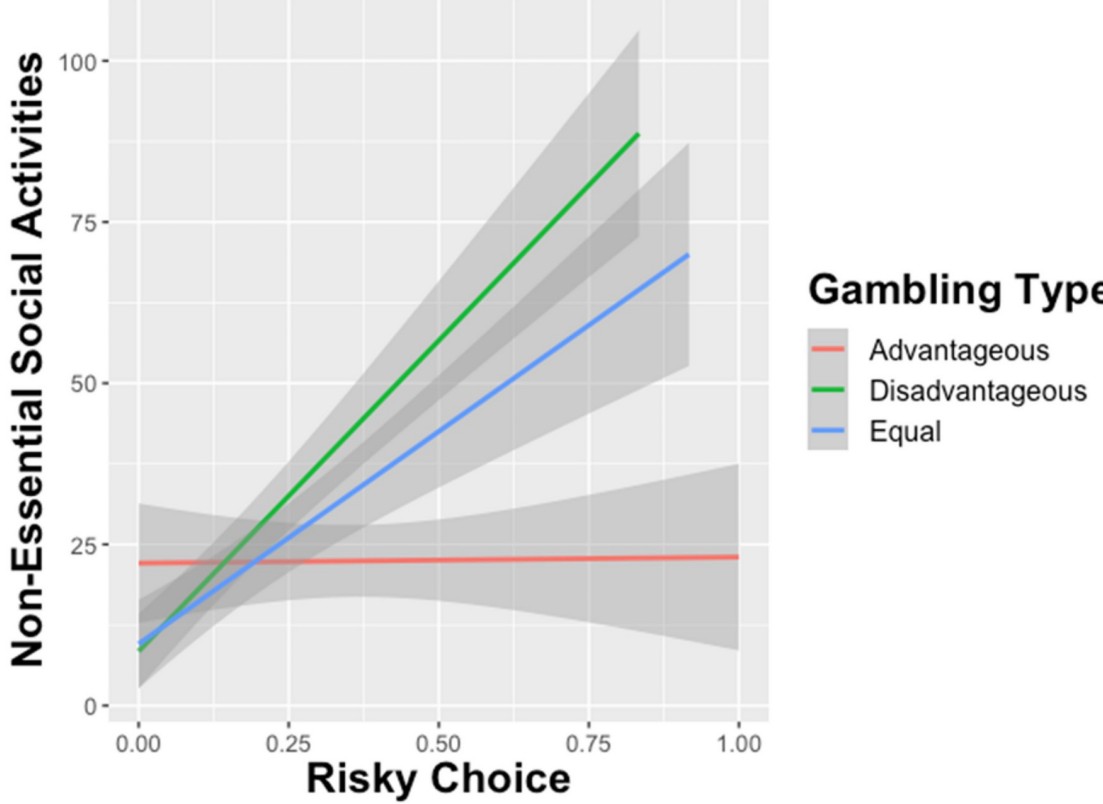

**Fig 3. Effect of risky choice on number of interpersonal social interactions by gambling type.** Advantageous gambles indicate that the expected value for the risky choice was higher than the safe gamble. Disadvantageous gambles indicate that the expected value for the risky option was lower than the safe option, and the expected values for both safe and risky choices were the same for Equal gambles. Higher proportion of Disadvantageous and Equal risky choices predicted greater engagement in non-essential social activities, an indicator of reduced social distancing behavior.

discounting nor difference in perceived risk were associated with the optimism bias. For the covariates included in the model, years of education (β = -.188, $p < .001$) predicted diminished optimism bias, or a decreased perception that others would be more likely to contract COVID-19 than them. No other predictors ($p$s>.10) or covariates were significant ($p$s>.05). The $R^2$ value for the overall model was.099.

**Perceived risk.**   Regression indicated that the proportion of risky choices (β = -0.200, $p < .001$) and temporal discounting (β = 0.104, $p = .035$) were predictive of Perceived Risk of engaging in activities in public settings. A difference also emerged for Political Affiliation (β = -0.270, $p < .001$) in which Democrats indicated a higher perceived risk of these activities compared to Republicans and Independents. No other covariates were significant ($p$s>.40). S7 Table in S1 File shows the regression results. The difference in perceived risk variable was not included as a predictor in this model as it was derived from this measure. The $R^2$ value for the overall model was.126.

## Path model

As an exploratory analysis, a path model was performed to examine the relationships between risky choice, temporal discounting, and perceived risk-taking on both actual social distancing and mask-wearing behavior together. We found an overall effect of risky choice on risk

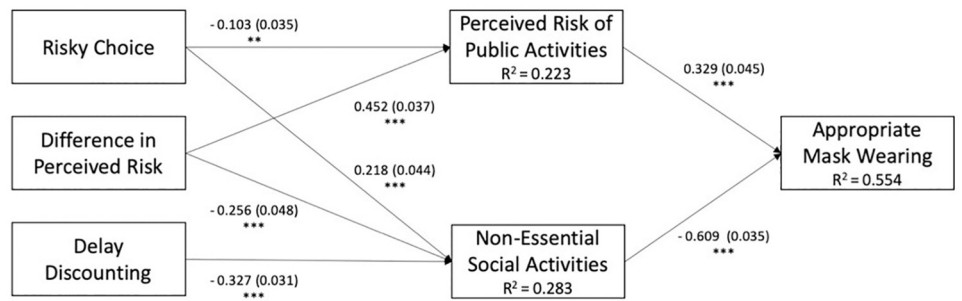

**Fig 4. Path model showing the relationship between risky choice, delay discounting, and appropriate mask wearing behavior.** Risky Choice indicates the overall proportion of risky decisions in the Risky Choice Task. The relationship between Delay Discounting scores and appropriate mask-wearing behavior was mediated by the frequency of engaging in non-essential social activities. The relationship between Risky Choice and appropriate mask-wearing behavior was mediated by both the frequency of engaging in non-essential social activities and one's perceived risk of engaging in public activities. Values in parentheses indicate standard error. ** indicates $p < .05$. *** indicates $p < .001$.

perceptions where mask-wearing is not being practiced. Participants who take more risky choices attribute less risk to going to places where people do not wear masks ($p = .003$). Furthermore, we found that the effect of risky choice on mask-wearing is mediated by perceived risk of public activities and engagement in non-essential social activities respondents who make more risky choices are more likely to attend to non-essential social activities ($p < .0001$) and perceive lower risk of engaging in public activities ($p < .05$). On the other hand, those who have higher levels of temporal discounting tend to avoid non-essential social activities ($p < 0001$). In terms of appropriate mask-wearing behaviors, those who attend non-essential social activities are less likely to wear masks ($p < .0001$), and those who attribute higher risks to public activities are more likely to wear masks ($p < .0001$). These effects are shown in Fig 4.

## Supplemental analyses

Self-reported risk-taking ($M = 36.53$, $SD = 24.15$) ranged from 0–100. There was a strong correlation between self-reported risk-taking with overall proportion of risky choices ($r = .53$, $p < .001$), appropriate mask-wearing ($r = -.37$, $p < .001$), interpersonal social interactions ($r = .35$, $p < .001$), and non-essential social activities ($r = .47$, $p < .001$). Thus, self-reported risk-taking also appears to have a strong relationship with COVID-19 preventative behavior like risky choices did. Moreover, one's level of worry about contracting COVID-19 was significantly related to the perceived risk of engaging in activities in public settings ($r = .52$, $p < .001$) and mask-wearing ($r = .18$, $p < .001$) but did not correlate with social distancing behavior. There was no association between stress-related uncertainty due to COVID-19 and risky choice, mask-wearing, or social distancing measures ($p$s>.50). The full correlational results are shown in S8 Table in S1 File. In terms of prosocial behavior, correlations between the Prosocial Behavioral Intentions Questionnaire and all outcome variables were nonsignificant, but there was an association between prosocial behavior on the Dictator Game and interpersonal social interactions ($r = .160$, $p = .025$) and non-essential social activities ($r = .196$, $p = .006$). Surprisingly, this association suggests that increased prosocial behavior on the Dictator Game was associated with less social distancing (S9 Table in S1 File). Supplemental exploratory analyses for political affiliation and a multivariate regression analysis with all dependent variables is shown in the (S10 Table in S1 File).

## Discussion

To mitigate the spread of COVID-19 transmission in the United States, the CDC and other government policy institutes have recommended that individuals wear masks in public places and practice social distancing [1]. Compliance with these recommendations has been inconsistent, and the pandemic remains a major public health crisis. The results of this study provide insight into how certain decision-making and motivation propensities are associated with compliance behaviors. The primary findings demonstrate that increased temporal discounting and risky decision-making are associated with less appropriate mask-wearing behavior and social distancing. These findings support our hypothesis.

The effect of risk-taking on COVID-19 compliance behavior was dependent on sensitivity to expected values. When a risky option has a higher expected value than a safe option, it is advantageous to choose the risky option. The results indicated that risky choices in these advantageous contexts were not predictive of compliance behavior. Instead, greater risk-taking in disadvantageous contexts in which the expected value for the risky option was lower than the safe option predicted diminished compliance behavior. This finding suggests that individuals who are less sensitive to changes in expected value and exhibit less adaptive risky decision-making behavior are less likely to engage in mask-wearing and social distancing during the pandemic. Individuals who chose the risky option more frequently in equal expected value contexts (when the expected values for the risky and safe options matched) behaved similarly to those who chose more disadvantageous risky choices. However, the magnitude of the relationship between risk-taking and noncompliant behavior was greater in the disadvantageous contexts than in the equal contexts. It should be noted that the average percentage of risky choices across all participants in equal and disadvantageous contexts was low—less than 20%. Moreover, over half (57%) of participants chose the safe option on all disadvantageous trials, and 47% chose the safe option on all equal expected value trials. The relationship between risk-taking and compliance behavior appears to be driven by a minority of participants, which may limit the generalizability of the findings. Nevertheless, the behavior of even a minority of individuals can have substantial impacts during a pandemic. Recent estimates indicate that approximately 10% of infected individuals account for 80% of COVID-19 transmission [75]. Though a minority of individuals engage in equal and disadvantageous risky decision-making, the corresponding decrease in compliance with mask-wearing and social distancing observed in this study could have wide-ranging consequences for spreading COVID-19.

Furthermore, previous work has shown that temporal discounting is associated with maladaptive health behaviors, including unhealthy eating and engaging in risky sexual activities [33–35]. Our findings build on this prior work by showing that temporal discounting is also predictive of engagement in COVID-19 preventative behaviors. Thus, this work provides empirical evidence that individual differences in risk-taking and motivation influence compliance with COVID-19 prevention guidelines.

Recent research on COVID-19 risk perception has observed that individuals tend to believe that others are more likely to contract COVID-19 than they are, which supports the optimism bias [45–47]. Our findings are consistent with this research; on average, individuals underestimated their likelihood of contracting COVID-19 compared to their peers by 7.9%. No relationship between risk-taking behavior or temporal discounting was associated with the magnitude of the optimism bias in this study. While it was anticipated that a stronger optimism bias would be associated with reduced compliance with COVID-19 preventative behavior, correlational results suggest an opposing view. The optimism bias was associated with increased social distancing and mask-wearing behavior, which was unexpected. However, concern for others has been associated with increased generosity towards strangers [76]. It is,

therefore, possible that increased risk perception of spreading COVID-19 to others may increase concern for others, leading to increased mask-wearing and social distancing to protect others. This explanation is speculative, and future research is needed to replicate this relationship and further examine the role of the optimism bias in COVID-19 preventative behaviors.

In addition to differences in risk perception of others versus self in contracting COVID-19, the relationship between perceived risk of engaging in public activities and COVID-19 preventative behaviors was examined. Risky decision-making and temporal discounting were not predictive of risk perception. However, the hypothesis that increased risk perception would be associated with greater social distancing and mask-wearing was supported through correlational results, which is consistent with other recent findings [40]. This observed relationship is also in line with prior work showing that when people perceive an event as high risk, they want the risk reduced and support the establishment of risk-reduction regulations [20, 21]. The desire for a risk to be reduced is not the same as taking action to reduce the risk yourself though, and it was unclear whether heightened risk perception would be associated with actual increases in COVID-19 preventative behavior. The study results shed light on this matter: heightened risk perception of engaging in public activities during the COVID-19 pandemic is associated with increased mask-wearing and social distancing.

This study further demonstrated that higher risk perception of public activities under nonsocial compared to social distancing conditions were predictive of greater mask-wearing and social distancing behavior. This finding suggests that individuals who feel that social distancing is effective are more likely to engage in such behaviors. From another perspective, however, this result may have implications for risk compensation, which proposes that individuals adapt their behavior based on their level of the perceived risk of that behavior, typically behaving in a more risk-taking way when perceived risk is low [77]. This theory has been observed in safety contexts in which having more protective measures in place, such as safety equipment [78, 79], increases people's risk-taking behavior because these protective measures decrease one's perceived level of risk. Applied to COVID-19 preventative behavior, it is possible that if people perceive the risk of social activities as lower when others are engaging in prophylactic behaviors—wearing masks or social distancing, then they may be more willing to empirically engage in those social activities. Future research is needed to examine risk compensation in the context of COVID-19 preventative behavior.

The relationship between several demographic factors and COVID-19 preventative behaviors were also examined in this study. Although prior H1N1 and COVID-19 research has observed age differences in virus risk perception [37, 41, 80], the present study found no significant relationship between age and preventative behaviors or risk perception of engaging in public activities during the COVID-19 pandemic. The results indicated that higher income levels predicted more compliance with mask-wearing guidelines but did not affect social distancing behavior. It was also expected that first-hand personal experience with COVID-19, including knowing someone who became sick or died, and negative financial consequences from the pandemic would lead to greater compliance behaviors. The results do not support these predictions; instead, both factors were associated with *reduced* social distancing. Further research is needed to examine why negative personal experience with COVID-19 may not necessarily lead to greater COVID-19 preventative behaviors.

Response to the pandemic has become highly politicized in the United States [3, 4]. This study showed that differences in political affiliation were predictive of perceived risk of engaging in public activities (assuming no social distancing), which is consistent with other recent findings [40]. Democrats reported a higher risk perception of engaging in public activities compared to Republicans. Exploratory analyses examining the effect of political affiliation alone on mask-wearing and social distancing showed that Democrats and Independents

engaged in more social distancing and appropriate mask-wearing behavior than Republicans (see S1 File). However, political affiliation was not significant in the regression model, which suggests that political affiliation is not predictive of mask-wearing and social distancing over and above the behavioral decision-making factors of risk-taking, temporal discounting, and perceived risk. In other words, risk-taking and temporal discounting are stronger predictors of mask-wearing and social distancing than political affiliation. Outside of political affiliation, the results indicated that those that participated in the study in December reported increased mask-wearing and social distancing than those that participated in September. This result is consistent with research from the COVID States Project [53] showing that mask-wearing behavior has increased since the pandemic began. Therefore, while political affiliation may be a divisive factor in terms of COVID-19 risk perception, the data suggest that, on the whole, individuals' mask-wearing and social distancing behavior increased between September and December 2020.

As supplementary analyses, the relationship between worry, risk perception, and COVID-19 prevention behaviors was also explored. The study results showed that although high levels of worry about contracting COVID-19 were associated with increased risk perception, worry was not associated with heightened compliance with COVID-19 prevention guidelines. Our supplementary results also showed that prosocial behavior was not associated with increased compliance with mask-wearing and social distancing in our sample. This finding diverges from previous research showing a positive relationship between worry and prosocial behavioral changes [36, 38, 42]. However, this result may suggest an interesting dissociation in which worry may exert differential effects on risk perception and actual risk behavior—people worried about contracting COVID-19 may perceive higher risks about engaging in public activities but still choose to engage in them anyway. In a self-destructive cycle, engaging in such perceived high-risk activities may then heighten worry about getting COVID-19 from those activities.

To explore the direction of the relationship between decision-making, risk perception, and COVID-19 preventative behaviors, a path analysis was performed. The results showed that risk perception and engagement in non-essential social activities mediated the relationship between decision-making and mask-wearing behavior. Specifically, individual differences in risky decision-making predicted both perceived risk and actual risk-reduction behavior. Risky decision-making predicts reduced risk perception which, in turn, leads to diminished mask-wearing behavior. One possible explanation for this relationship is that individuals who regularly make risky choices may not view the COVID-19 pandemic as perilous relative to some of the other risks they have taken. The path model has demonstrated that the relationship between temporal discounting and mask-wearing behavior is mediated by engagement in non-essential social activities, but not risk perception. People who are more motivated by immediate gratification engage in more non-essential social interactions to experience the pleasure of those activities and interactions at the expense of long-term public health consequences. Engaging in more of these immediately rewarding social activities may mean that these individuals encounter more incidents in which masks and social distancing *should* be employed, yet they choose to engage in less appropriate mask-wearing behavior. Risky decision-making behavior, temporal discounting, and risk perception collectively predicted 55% of the variance in appropriate mask-wearing behavior. Individual differences in general decision-making patterns are therefore highly predictive of who complies with COVID-19 prevention guidelines.

## Implications

Despite widespread feelings of anxiety and fear surrounding COVID-19, many individuals still behave in a way that is inconsistent with social distancing guidelines. Some individuals who

report lower adherence to preventive measures may have been disregarding such guidelines throughout the entire pandemic duration. However, others may have been rigidly following guidelines early in the pandemic but may have begun to disregard the warnings due to a type of "quarantine burnout" in which people have simply run out of willpower to enact safe practices every time they leave their house. This idea fits with our finding that instant gratification may influence COVID-19 compliance behaviors. While people may still believe that COVID-19 is a serious risk, the value of long-term health and engaging in COVID-19 preventative behaviors may decline as the time duration of the pandemic increases.

## Limitations and future directions

This study is not without limitations. Firstly, all study participants were living in the United States, an individualistic culture in which the needs of the individual are often valued over the needs of the community. Individualistic cultures may be more susceptible to allowing preferences for short-term rewards like socializing over long-benefits of community health to guide compliance behavior. As previous work has shown that responses to the coronavirus vary across countries [40], the present results may not extend to other countries or cultures with more collectivist values.

A limitation in our design is that overall knowledge regarding coronavirus information was not assessed, which may affect COVID-19 preventative behavior. Moreover, as with many studies that utilize self-report measures, social desirability biases may have influenced participants' risky choices or disclosure of mask-wearing and social distancing behavior. This study was also conducted online, and as such our findings may not generalize to individuals with limited internet or computer access. The online nature of the study means that experimental control of the study environment was limited. We also note that other factors outside of this study including personality [43], line of work, mental health, and emotional states [42] may also play a role in COVID-19 preventative behavior. These other unexamined factors may covary with risk-taking or temporal discounting, which has the potential to inflate the observed estimation of effect sizes observed in this study. We therefore caution that the study effect sizes may represent an upper bound of the overall effect of risky decision-making and temporal discounting on COVID-19 preventative behavior. Moreover, it should be emphasized that the observed effects in this study show associations, rather than causal effects.

Furthermore, according to the risk-as-feelings hypothesis, anticipatory 'in-the-moment' emotions have a substantial impact on one's decisions [81]. The risk-as-feelings hypothesis can explain decisions that diverge from ones that may objectively seem to be the best course of action. However, our study design did not capture feelings about COVID-19 or anticipatory emotions, and future research should consider examining the effect of decision-making patterns on COVID-19 preventative behavior using a risk-as-feelings framework.

## Conclusions

The effects of the COVID-19 pandemic have been widespread and deleterious. This work sought to characterize the role of risky decision-making and motivation factors that underlie COVID-19 preventative behavior. This study provides empirical evidence that increased risk-taking during decision-making, diminished risk perception, and motivation for immediate over delayed gratification predict *reduced* adherence to COVID-19 prevention guidelines. This information may provide insight into ways to increase compliance with these guidelines. While public health guidelines and messaging need to emphasize the risks of COVID-19, it is also important to convey activities and opportunities that can be immediately rewarding. This

approach may provide an appropriate outlet for those with more risk-taking tendencies and those who seek out immediate rewards.

## Supporting information

**S1 File.**
(DOCX)

## Author Contributions

**Conceptualization:** Kaileigh A. Byrne.

**Data curation:** Kaileigh A. Byrne, Reza Ghaiumy Anaraky.

**Formal analysis:** Kaileigh A. Byrne, Reza Ghaiumy Anaraky.

**Funding acquisition:** Kaileigh A. Byrne.

**Investigation:** Kaileigh A. Byrne.

**Methodology:** Kaileigh A. Byrne.

**Project administration:** Kaileigh A. Byrne.

**Resources:** Kaileigh A. Byrne.

**Supervision:** Kaileigh A. Byrne.

**Visualization:** Reza Ghaiumy Anaraky.

**Writing – original draft:** Kaileigh A. Byrne, Stephanie G. Six, Reza Ghaiumy Anaraky, Maggie W. Harris, Emma L. Winterlind.

**Writing – review & editing:** Kaileigh A. Byrne.

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
