## [Decision Letter · Decision Letter 0]

29 Dec 2020

PONE-D-20-32678

Risk-Taking Unmasked: Using Risky Choice and Temporal Discounting to Explain COVID-19 Preventative Behaviors

PLOS ONE

Dear Dr. Byrne,

Thank you for submitting your manuscript to PLOS ONE. After careful consideration, we feel that it has merit but does not fully meet PLOS ONE’s publication criteria as it currently stands. Therefore, we invite you to submit a revised version of the manuscript that addresses the points raised during the review process.

Both reports are attached. As you will see the referees are asking for clarifications (for instance, the measure of risk aversion). Both referees are concerned about the sample size (and power), in particular with the n=20 sample of Clemson. Perhaps you may consider enlarge the sample size. This is a personal recommendation only. I am not asking for new experiments.

Both referees are mention that the entire experiment if self-reported (no incentives). I am in favor of hypothetical experiments and in fact I do experiments comparing  hypothetical and incentivised (as both referees cite)... but you need to explain this. Its important to clarify and show the potential limitations.

There are many other comments in the report you need to handle. Please keep in mind that I will send back the paper to the very same referees.

We look forward to receiving your revised manuscript.

Kind regards,

Pablo Brañas-Garza, PhD Economics

Academic Editor

PLOS ONE

Journal Requirements:

Reviewers' comments:

Reviewer's Responses to Questions

**Comments to the Author**

1. Is the manuscript technically sound, and do the data support the conclusions?

Reviewer #1: Yes

Reviewer #2: Partly

2. Has the statistical analysis been performed appropriately and rigorously? 

Reviewer #1: Yes

Reviewer #2: N/A

3. Have the authors made all data underlying the findings in their manuscript fully available?

Reviewer #1: Yes

Reviewer #2: No

4. Is the manuscript presented in an intelligible fashion and written in standard English?

Reviewer #1: Yes

Reviewer #2: No

5. Review Comments to the Author

Reviewer #1: The general idea of the paper is to analyze the correlation of risk preferences, temporal discounting, risk perception and measures of appropriate mask wearing, social distancing.

In order to do the analysis, the authors run an online experiment (n=225). Participants were recruited using MTurk (N=220) and the undergraduate subject pool at Clemson University (N=20).

The work is very well written and yields interesting results on the relations between different behaviors measures and the proper use of masks and social distancing. Despite this, it is a correlational study and some results should be considered with caution.

General comments

The paper analyzes the relationship between COVID-19 preventative behaviors and individual differences in four classic judgment and decision-making constructs. But, the correct use of masks and compliance with social distancing can be seen also as a collective action problem, where there are other hypotheses that can explained how and why people cooperates. Also, the COVID could have a direct impact on risk, delayed discounting and selfishness (see Brañas et al., 2020a; Adena and Harke, 2020). At the end, the results that authors can be seen is that people became more selfish, impatient or risk averse as a response to this situation, and they become even more as the day passed in the time window that did the survey. Probably, author need to add a paragraph with this discussion and might controlled for days fixed effect in the regression analysis.

Also, the independent variables used do not reflect directly compliance with COVID-19 prevention guidelines. Preventive measures are public knowledge, so many people can answer what is socially desirable and do not necessarily reveal their true intention. Probably authors need to discuss the social desirability bias in their hypotheses.

Specific comments

a) They talked about the study’s limitations; they need to analyze how representative is the sample to the standard US population. With 225 observations, probably is not representative and the external validity of the results is very restricted.

b) Despite the power calculations made, the number of observations is low for a Mturk sample. However, the design is very good and the results are very interesting, so authors should think about redoing the experiment with a larger sample. It is not necessary to do it in Mturk. Jorrat (2020) suggests a procedure to do online experiments in a short time and achieve a high number of observations.

c) Another interesting independent variable to analyze could be the difference between the perceived risk of the different activities with and without social distancing. This could be a measure of how effective people think social distancing is.

d) Authors need to discuss about why hypothetical time a risk experimental measures are a good proxy of incentivized ones. These papers study this experimental question:

Brañas-Garza, P., Jorrat, D., Espín, A. M., & Sanchez, A. (2020). Paid and hypothetical time preferences are the same: Lab, field and online evidence. arXiv preprint arXiv:2010.09262.

Brañas-Garza, P., Estepa Mohedano, L., Jorrat, D., Orozco, V., & Rascon-Ramirez, E. (2020). To pay or not to pay: Measuring risk preferences in lab and field.

Falk, A., Becker, A., Dohmen, T. J., Huffman, D., and Sunde, U. (2015). The preference survey module: A validated instrument for measuring risk, time, and social preferences. IZA Discussion Paper.

e) A regression analyses with all the dependent variables is need it. Authors can made different specifications and add each of the four variables separately and other specifications with all the variables. Authors also need to put the regressions tables in the supplementary materials.

References:

Adena, M. & Harke, J, (2020). COVID-19 and pro-sociality: the effect of pandemic severity and increased pandemic awareness on charitable giving. Mimeo.

Branas-Garza, P., Jorrat, D. A., Alfonso, A., Espin, A. M., García, T., & Kovarik, J. (2020). Exposure to the Covid-19 pandemic and generosity. https://doi.org/10.31234/osf.io/6ktuz

Jorrat, D. A. (2020). Recruiting experimental subjects using WhatsApp. https://doi.org/10.31234/osf.io/6vgec

Reviewer #2: The article aims to find evidence supporting the idea that individual behavior to prevent the diffusion of COVID19, or attitudes toward the riskiness of the pandemic, are linked to individual underlying preferences, such as in particular risk aversion, risk perception, time discounting. Preventative individual behaviors that are considered are wearing facemasks, avoiding large gatherings. Attitudes are perception of risk and optimism bias. The authors find evidence of the relevance of risk preferences on such dependent variables.

Major points:

1) Two categories of individual motivations are relevant in preventative behavior. One category refers to risk aversion, and this is considered by the authors. Another concerns pro-sociality. For instance, wearing masks is at the same time something that protects the individual from the infection, but also protects others from catching the infection (e.g. Cheng et al., 2020). The participants themselves are aware of this aspect, as apparent from responses to an item of the questionnaire. For this reason, I find the design of the study incomplete, because it does not include a measure for pro-sociality. If pro-sociality was positively correlated with risk-aversion, then risk aversion would arguably pick up some of the effects of pro-sociality, thus inflating the effect size. Since there does not seem to be items measuring pro-sociality in the questionnaire, this appears an irreparable flaw of the design. The authors should at least discuss the extent to which their estimation of the effect of risk aversion is an upper bound of the real effect.

2) I am puzzled by the measurement of risk preferences. The authors divide lotteries into three types – risk advantageous, risk disadvantageous, and equal risk, but only find significant effects for the latter two. It is not theoretically clear why this should be the case, and why we should consider these three types of lotteries separately from each other. Individuals who would prefer lotteries to the safe option, when lotteries are disadvantageous (or “equal”, in the authors’ wording), are normally referred to as “risk lovers”, or “risk-neutral” individuals. In my knowledge, risk lovers and risk neutrals are a minority of the population, while most people are “risk averse”. Since the authors only find significant effects for “disadvantageous” or “equal” lotteries, I wonder whether this effect is only driven by a relative minority of the sample. This would not be an uninteresting result per se, but there may be issues of generalizability. I would have liked to see simple descriptive statistics on this variable, but they were not reported.

3) Another related point concerns the construction of the risk aversion variable. Dividing lotteries into these three levels and measuring the percentage of risky choices within each level seems a rather coarse approach. First, within each level, different lotteries will have different expected payoff values and thus different degrees of “advantageousness” and “disadvantageousness”. The level of risk is fixed in equal risky choice, but here (presumably), the size of the pie was manipulated. Hence, considerable information seems to have been ignored when constructing the indexes. The approach that I would advice is instead different. Drawing on contributions in experimental economics, a single parameter for individual risk aversion may be estimated, on the basis of choices throughout the three levels of risk. An approach models utility as “Constant Relative Risk Aversion”, and the curvature of the utility function (which is given by one parameter) is a synthetic indicator of an individual’s risk aversion (see Harrison & Rutström, 2008, Wakker, 2008). More sophisticated calibrations are also possible, including in particular the estimation of a loss aversion parameter (Abdellaoui et al., 2008). Incidentally, it is not clear from the text whether lotteries including losses were administered, but this seems to be the case from the examples reported in the questionnaire registred at OSF. Other approaches would be possible, but the current approach is unsatisfactory as it stands at the moment, in my view.

4) The authors use linear regressions, but given the discrete nature of the dependent variable, an ordered logit model, or interval regression, would have been more appropriate. I am also not clear why authors use repeated measures for the risky hypothetical choices, done over the three levels (advantageous, equal, disadvantageous) – which, incidentally, was not part of the pre-analysis plan. This seems to arbitrarily inflate the power of the risky decision variable, and does not appear to be grounded in theory. Tables with the regression results should be reported, either in the main text or the Appendix.

5) At page 16 the authors state: “equal gambles (N=12) in which the expected value for the risky and sure options were identical or nearly identical […]”. It appears arbitrary to classify lotteries whose expected value is the same or “nearly the same” as the certain option as belonging to the same category. In this sense, having 12 different lotteries that are “equal” seem to be rather excessive. I understand this may be customary in the strand of literature the authors are following. If so, this aspect should be clarified.

6) Page 16-17: “The Appendix shows the full list of questions”. I could only find three questions in the OSF website.

7) An obvious concern is that all the variables are self-reported. In particular, it was not clear to me that the choice of risky lotteries had not been monetarily incentivized. Only in the pre-registration of hypothesis I could eventually find this information. To the very least, the authors should discuss the implications of hypothetical Vs. monetarily incentivized questions (e.g. Beattie & Loomes, 1997; Brañas-Garza et al., 2020; Carlsson & Martinsson, 2001; Donkers et al. 2001).

8) I appreciate that authors followed the good practice of pre-registering their hypotheses. It is clear that the analysis reported in the paper generally followed the pre-analysis plan. Nevertheless, there appear to be some deviations, and these should be flagged out in the paper. In particular, (a) the second hypothesis (Greater stress-related uncertainty due to COVID-19 will be associated with decreased risk-taking) has not been analyzed in the paper. (b) Dependent variable (3), “Willingness to return to work (continuous scale)” has not been analyzed, while “Perceived Risk” has been. I am very much against the pre-registration acting as a straitjacket on what authors should report in the paper. But transparency requires to inform the reader as to why some modifications of the pre-analysis plan were undertaken, what is post-hoc rationalization of results, and what is post-diction rather than pre-diction.

9) The authors may benefit from including in their analysis the report issued by “The Covid States Project” (https://covidstates.org/), led by Northeastern University. In particular, the report on COVID19-associated behavior signals that wearing face masks is the only behavior that, among those considered, has been on the rise, while others, like social distancing, have been declining since measuring began (at the end of April). See: https://covidstates.org/
https://kateto.net/covid19/COVID19%20CONSORTIUM%20REPORT%2026%20BEHAVIOR%20NOV%202020.

Minor points:

10) The authors give the impression to associate “rationality” with decisions taken according to Expected Utility Theory (page 5). This is incorrect, and it does not seem to be necessary, also in the light of the authors’ following discussion. I do not see the need to incorporate the discussion of prospect theory in the introduction, as this theory is not used later on.

11) It is really odd that a small portion of the sample (N=20) is made up of Clemson university students.

12) I find the paper too wordy in the introduction and discussion. There is no need to motivate the scope of the paper multiple times, or to expand theoretical review much beyond what is actually used in the paper. The policy implication discussed in the discussion to send out “positive messages” of what people may do, rather than negative messages of what people can’t do, does not seem to be supported by evidence produced in this paper.

13) There are several inaccuracies: Page 5: “Thus, the probabilities of COVID-19 infection rates are known”. Strictly speaking, this is not true, because the actual cases are arguably much higher, and unknown, than reported cases. The modalities of infection are still not fully known.

14) The author states “It is unclear how perceived risk will influence actual COVID-19 preventative behavior” (Page 6) and “It is unclear how individual differences in temporal discounting relate to COVID-19 preventative behaviors”. (Page 7). But at page 10 they say they hypothesize that higher perceived risk and higher temporal discounting are linked with lower preventative behavior (as should reasonably be expected).

15) Page 9: “Because there is currently no vaccine available”: This should obviously now be updated.

16) In the questionnaire text reported in the OSF website, the last question reads: “There’s a 75% chance that you will lose $100, but a $25 chance that you will not lose any money.” I hope the typo was amended when presented to participants.

17) Page 19: Truncated sentence: “To examine whether the optimism bias, a paired samples t-test was conducted”.

18) Page 11: The discussion of the quality of M-Turk samples is presented before saying that the sample was from M-Turk.

19) Page 2: the authors claim that the sample is “representative”, but with N=225 this can never be the case.

References

Abdellaoui, M., Bleichrodt, H., & l’Haridon, O. (2008). A tractable method to measure utility and loss aversion under prospect theory. Journal of Risk and uncertainty, 36(3), 245.

Beattie, J., & Loomes, G. (1997). The impact of incentives upon risky choice experiments. Journal of Risk and Uncertainty, 14(2), 155-168.

Brañas-Garza, P., Jorrat, D., Espín, A. M., & Sanchez, A. (2020). Paid and hypothetical time preferences are the same: Lab, field and online evidence. arXiv preprint arXiv:2010.09262.

Carlsson, F., & Martinsson, P. (2001). Do hypothetical and actual marginal willingness to pay differ in choice experiments?: Application to the valuation of the environment. Journal of Environmental Economics and Management, 41(2), 179-192.

Cheng, Vincent CC, Shuk-Ching Wong, Vivien WM Chuang, Simon YC So, Jonathan HK Chen, Siddharth Sridhar, Kelvin KW To et al. "The role of community-wide wearing of face mask for control of coronavirus disease 2019 (COVID-19) epidemic due to SARS-CoV-2." Journal of Infection (2020).

Donkers, B., Melenberg, B., & Van Soest, A. (2001). Estimating risk attitudes using lotteries: A large sample approach. Journal of Risk and uncertainty, 22(2), 165-195.

Harrison, G. W., & Rutström, E. E. (2008). Risk aversion in the laboratory. Research in experimental economics, 12(8), 41-196.

Wakker, P. P. (2008). Explaining the characteristics of the power (CRRA) utility family. Health economics, 17(12), 1329-1344.

6. PLOS authors have the option to publish the peer review history of their article (what does this mean?). If published, this will include your full peer review and any attached files.

Reviewer #1: No

Reviewer #2: No

---

## [Author Response · Author response to Decision Letter 0]

13 Jan 2021

Response to Reviewer Comments

Reviewer #1: The general idea of the paper is to analyze the correlation of risk preferences, temporal discounting, risk perception and measures of appropriate mask wearing, social distancing. In order to do the analysis, the authors run an online experiment (n=225). Participants were recruited using MTurk (N=220) and the undergraduate subject pool at Clemson University (N=20). The work is very well written and yields interesting results on the relations between different behaviors measures and the proper use of masks and social distancing. Despite this, it is a correlational study and some results should be considered with caution.

General comments

The paper analyzes the relationship between COVID-19 preventative behaviors and individual differences in four classic judgment and decision-making constructs. But, the correct use of masks and compliance with social distancing can be seen also as a collective action problem, where there are other hypotheses that can explained how and why people cooperates. Also, the COVID could have a direct impact on risk, delayed discounting and selfishness (see Brañas et al., 2020a; Adena and Harke, 2020). At the end, the results that authors can be seen is that people became more selfish, impatient or risk averse as a response to this situation, and they become even more as the day passed in the time window that did the survey. Probably, author need to add a paragraph with this discussion and might controlled for days fixed effect in the regression analysis.

We certainly agree that there are likely other factors and hypotheses that could affect how and why people cooperate with COVID-19 prevention guidelines. It was not our intention to provide a comprehension examination of how all plausible decision-making factors may influence COVID-19 prevention guidelines. We have made efforts to clarify this and temper our research objectives throughout the Introduction. For example, we have now replaced the prior reference to the study being a “systematic empirical research” endeavor with a statement that “empirical research examining decision-making factors that influence compliance with mask-wearing and social distancing guidelines is lacking. Therefore, in addition to demographic factors, this study seeks to examine whether certain decision-making constructs, such as general risk-taking propensity and temporal discounting, are predictive of compliance with appropriate mask-wearing and social distancing behaviors”. 

Additionally, in the Limitations section of the Discussion, we state that “we did not include an exhaustive list of possible influences on compliance with COVID-19 prevention guidelines in this study. Other factors including pro-sociality (Brañas-Garza et al., 2020), personality (Nofal et al., 2020), line of work, mental health, and emotional states (Harper et al., 2020) may also play a role in COVID-19 preventative behavior”. 

With regard to controlling for days, we note that Wave 1 (Sept 7 -11) and Wave 2 (Dec 29 – 30) were collected over a very narrow range of days. However, it is certainly possible that participants’ social distancing and mask-wearing behavior in September may differ from behavior in December. Consequently, we now include Wave (September vs. December) as a fixed effect in the regression analyses. 

Also, the independent variables used do not reflect directly compliance with COVID-19 prevention guidelines. Preventive measures are public knowledge, so many people can answer what is socially desirable and do not necessarily reveal their true intention. Probably authors need to discuss the social desirability bias in their hypotheses.

The dependent variables (appropriate mask-wearing, avoiding nonessential indoor spaces, and avoiding in-person gatherings) are direct prevention recommendations from the CDC. 

The point that people may think that compliance with COVID-19 preventative measures is socially desirable is debatable; because the pandemic is highly politicized in the U.S., many people proudly refuse to wear masks or social distance. In general, social desirability biases are an artifact of the majority of psychological research studies involving self-report. To address this issue, we added the following sentence to the Limitations section of the Discussion: “Moreover, as with many studies that utilize self-report measures, social desirability biases may have influenced participants’ risky choices or disclosure of mask-wearing and social distancing behavior”. 

Specific comments

a) They talked about the study’s limitations; they need to analyze how representative is the sample to the standard US population. With 225 observations, probably is not representative and the external validity of the results is very restricted.

The updated sample population has fewer African American and Hispanic individuals than the US population, so we have now removed the reference to the sample being representative. 

b) Despite the power calculations made, the number of observations is low for a Mturk sample. However, the design is very good and the results are very interesting, so authors should think about redoing the experiment with a larger sample. It is not necessary to do it in Mturk. Jorrat (2020) suggests a procedure to do online experiments in a short time and achieve a high number of observations. 

We have now recruited an additional 200 participants through MTurk. Additionally, we removed the 20 participants from the Clemson sample per Reviewer 2’s suggestions. The current sample size (N=404) is now nearly double the original sample size. Given the power analysis results, exceeding this sample size by more than double may artificially inflate the observed study effects given that it is easier to reach statistical significance with larger samples. Therefore, the present sample size should now be sufficient to appropriately address the study research questions while minimizing problems associated with both over- and under-powered studies. 

c) Another interesting independent variable to analyze could be the difference between the perceived risk of the different activities with and without social distancing. This could be a measure of how effective people think social distancing is.

Although not originally in our hypotheses or design, we agree that examining the difference in the perceived risk with and without social distancing is a valuable contribution to the study. Not only could this measure provide insight on the perceived effectiveness of social distancing, but it also has implications for risk compensation behavior in the context of the pandemic. To incorporate this addition, we have made several modifications to the manuscript. 

First, we have added this measure (Perceived Risk Difference) to the Method section, to each of the regression results, and to the path model analysis. The results show that greater perceived risk of activities when not socially compared to when people are socially distancing was associated with greater mask-wearing behavior and social distancing. 

Finally, we added a paragraph describing the implications of this result in the Discussion. In particular, we state, “This study further demonstrated that higher risk perception of public activities under non-social compared to social distancing conditions were predictive of greater mask-wearing and social distancing behavior. This finding suggests that individuals who feel that social distancing is effective are more likely to engage in such behaviors. From another perspective, however, this result may have implications for risk compensation, which proposes that individuals adapt their behavior based on their level of perceived risk of that behavior, typically behaving in a more risk-taking way when perceived risk is low (Wilde, 1982). This theory has been observed in safety contexts in which having more protective measures in place, such as safety equipment (e.g., Hansanzadeh et al., 2020; Hedlund, 2000), increases people’s risk-taking behavior because these protective measures decrease one’s perceived level of risk. Applied to COVID-19 preventative behavior, it is possible that if people perceive the risk of social activities as lower when others are engaging in prophylactic behaviors—wearing masks or social distancing, then they may be more willing to engage in those social activities. Future research is needed to empirically examine risk compensation in the context of COVID-19 preventative behavior”. 

d) Authors need to discuss about why hypothetical time a risk experimental measures are a good proxy of incentivized ones. These papers study this experimental question:

Brañas-Garza, P., Jorrat, D., Espín, A. M., & Sanchez, A. (2020). Paid and hypothetical time preferences are the same: Lab, field and online evidence. arXiv preprint arXiv:2010.09262.

Brañas-Garza, P., Estepa Mohedano, L., Jorrat, D., Orozco, V., & Rascon-Ramirez, E. (2020). To pay or not to pay: Measuring risk preferences in lab and field.

Falk, A., Becker, A., Dohmen, T. J., Huffman, D., and Sunde, U. (2015). The preference survey module: A validated instrument for measuring risk, time, and social preferences. IZA Discussion Paper.

We agree that this distinction should have been clarified in the original manuscript. We now address research on real vs. hypothetical rewards in temporal discounting in the Temporal Discounting sub-section of the Methods by stating, “A significant body of research comparing the effects of real to hypothetical rewards has demonstrated that temporal discounting rates are highly similar under both conditions (e.g., Brañas-Garza et al., 2020c; Johnson & Bickel, 2002; Locey, Jones, & Rachlin, 2011; Madden et al., 2004), which suggests that hypothetical rewards are a valid proxy for incentivized rewards in temporal discounting experiments.”

Additionally, we discuss this distinction for risk-taking in the Risky Choice Task sub-section of the Methods by stating, “Most recent studies have shown that decisions on risky choice tasks are not significantly altered under hypothetical compared to real reward conditions (e.g., Brañas-Garza et al., 2020a; Carlsson & Martinsson, 2001; Hinvest & Anderson, 2010; Wiseman & Levin, 1996), though some exceptions have been observed (Slovic, 1969)”. 

These additions serve to clarify why hypothetical temporal and risky choice experimental measures are valid proxies of incentivized ones. 

e) A regression analyses with all the dependent variables is need it. Authors can made different specifications and add each of the four variables separately and other specifications with all the variables. Authors also need to put the regressions tables in the supplementary materials.

The path analysis has all the dependent variables, and path analysis is an extension of regression. Although the path analysis has redundancies with a multivariate regression analysis, we added the results of the multivariate regression to the Supplementary Material (Table S9). The tables with the regression results are also now reported in the Supplementary Material. 

References:

Adena, M. & Harke, J, (2020). COVID-19 and pro-sociality: the effect of pandemic severity and increased pandemic awareness on charitable giving. Mimeo.

Branas-Garza, P., Jorrat, D. A., Alfonso, A., Espin, A. M., García, T., & Kovarik, J. (2020). Exposure to the Covid-19 pandemic and generosity. https://doi.org/10.31234/osf.io/6ktuz

We have added the Branas-Garza et al., 2020 citation to the manuscript. Mimeo appears to be a private repository; we requested access to the Adena & Harke, 2020 preprint, but Mimeo did not provide it. However, we do have other citations (e.g., Dryhurst et al., 2020) that address the relationship between prosocial behavior and COVID-19. 

Reviewer #2: The article aims to find evidence supporting the idea that individual behavior to prevent the diffusion of COVID19, or attitudes toward the riskiness of the pandemic, are linked to individual underlying preferences, such as in particular risk aversion, risk perception, time discounting. Preventative individual behaviors that are considered are wearing facemasks, avoiding large gatherings. Attitudes are perception of risk and optimism bias. The authors find evidence of the relevance of risk preferences on such dependent variables.

Major points:

1) Two categories of individual motivations are relevant in preventative behavior. One category refers to risk aversion, and this is considered by the authors. Another concerns pro-sociality. For instance, wearing masks is at the same time something that protects the individual from the infection, but also protects others from catching the infection (e.g. Cheng et al., 2020). The participants themselves are aware of this aspect, as apparent from responses to an item of the questionnaire. For this reason, I find the design of the study incomplete, because it does not include a measure for pro-sociality. If pro-sociality was positively correlated with risk-aversion, then risk aversion would arguably pick up some of the effects of pro-sociality, thus inflating the effect size. Since there does not seem to be items measuring pro-sociality in the questionnaire, this appears an irreparable flaw of the design. The authors should at least discuss the extent to which their estimation of the effect of risk aversion is an upper bound of the real effect.

In the new second wave of data collection, we added two measures of pro-sociality: (1) the Prosocial Behavioral Intentions Scale (Baumsteiger & Siegal, 2019) and (2) a version of the Dictator Game. Neither the Prosocial Behavioral Intentions score (r = .047, p = .516) nor the Dictator Game measure (r = .058, p = .421) were correlated with overall risk-taking in the sample. The null correlation should address the reviewer’s concern that multicollinearity between risk-taking/risk aversion and pro-sociality could have inflated the effect size for risk-taking/risk aversion. 

We note that the correlations between the Prosocial Behavioral Intentions Questionnaire and all outcome variables were nonsignificant, but there was an association between Dictator Game pro-social behavior and interpersonal social interactions (r = .160, p=.025) and non-essential social activities (r = .196, p = .006). In other words, those that were more prosocial in the Dictator Game (i.e., gave more to the receiver than oneself) reported more social interactions and engaging in more-essential activities more (i.e., less social distancing). We considered adding these results regarding pro-sociality to the manuscript, but we feel that it muddies the waters and decreases the clarity of the manuscript and findings. 

Correlations between Prosocial Behavioral Intentions Questionnaire, Dictator Game, and Study Outcome Measures

 Appropriate Mask Wearing Interpersonal Social Interactions Social Activities Perceived Risk Optimism Bias

PBIS 0.053 0.046 -0.005 -0.036 -0.133

Dictator Game -0.087 0.16* 0.196* 0.042 -0.094

Note. PBIS = Prosocial Behavioral Intentions Scale. The Dictator Game is defined as the amount participants opted to give to the received minus the amount kept for oneself. Higher scores reflect greater pro-sociality. 

* indicates p<.05

We would also like to note that our objective in this study was not to comprehensively consider all possible decision-making constructs that may relate to COVID-19 preventative behavior. There are likely other constructs outside of pro-sociality that may contribute to COVID-19 preventative behavior as well, but these are outside the scope of this investigation. The original manuscript may have alluded that the study was trying to examine all decision-making or motivational factors that may affect COVID-19 preventative behavior. To address this issue, we made several edits throughout the Introduction to ensure that the objectives of the study are not overstated. 

Furthermore, in the Limitations section of the Discussion in which we state that “we did not include an exhaustive list of possible influences on compliance with COVID-19 prevention guidelines in this study,” we have now added pro-sociality to the list of other factors that may play a role in COVID-19 preventative behavior. We further elaborate that “these other unexamined factors may covary with risk-taking or temporal discounting, which has the potential to inflate the observed estimation of effect sizes observed in this study. We therefore caution that the study effect sizes may represent an upper bound of the overall effect of risky decision-making and temporal discounting on COVID-19 preventative behavior”. 

Reference:

Baumsteiger, R., & Siegel, J. T. (2019). Measuring prosociality: The development of a prosocial 

behavioral intentions scale. Journal of Personality Assessment, 101(3), 305-314.

Dictator Game Instructions: 

Imagine that a store is having a grand opening and is giving different amounts of money between $5 and $200 cash to the first 100 shoppers. You are shopper #99, and you get $100 cash.

Unfortunately, the store employee miscounted, and they do not have any money to give to shopper #100. The employee asks you if you would like to give some of your $100 to shopper #100. You do not know shopper #100, and it is very unlikely you will ever see them again in the future. How much money (if any) would you leave for shopper #100?

The outcome measure was defined as the amount participants opted to give to shopper #100 minus the amount they opted to keep for themselves. Higher scores reflect greater pro-sociality.

2) I am puzzled by the measurement of risk preferences. The authors divide lotteries into three types – risk advantageous, risk disadvantageous, and equal risk, but only find significant effects for the latter two. It is not theoretically clear why this should be the case, and why we should consider these three types of lotteries separately from each other. Individuals who would prefer lotteries to the safe option, when lotteries are disadvantageous (or “equal”, in the authors’ wording), are normally referred to as “risk lovers”, or “risk-neutral” individuals. In my knowledge, risk lovers and risk neutrals are a minority of the population, while most people are “risk averse”. Since the authors only find significant effects for “disadvantageous” or “equal” lotteries, I wonder whether this effect is only driven by a relative minority of the sample. This would not be an uninteresting result per se, but there may be issues of generalizability. I would have liked to see simple descriptive statistics on this variable, but they were not reported.

To clarify the measurement of risk preferences, we have added further information to the Methods section (p. 15 – 16). Importantly, we now state that the risky choice task is similar to the cups task and that this analytical approach is in line with previous research with this task and other related tasks.

We understand that the theoretical information motivating this approach of risky choices was lacking from the original manuscript. We have added a paragraph to the Introduction (p. 4 bottom – p. 5) to provide background information on why it is important to distinguish between risk-taking and expected values. For example, we state that sensitivity to expected utility can provide objective information about adaptive decision-making performance (Weller, Levin, & Bechara, 2010). Choosing risky options with higher expected utility maximizes is economically advantageous, while choosing risky options that have a lower expected utility than a safe option is maladaptive and can often lead to sub-optimal decision-making (Weller, Levin, & Bechara, 2010; von Neumann & Morgenstern, 1944). 

In the Discussion, we have added a paragraph (p. 28, bottom – p. 29) to explain why examining both risk level and expected values may have important implications. For example, we state that “This finding suggests that individuals who are less sensitive to changes in expected value and exhibit less adaptive risky decision-making behavior are less likely to engage in mask-wearing and social distancing during the pandemic. Individuals who chose the risky option more frequently in equal expected value contexts (when the expected values for the risky and safe options matched) behaved similarly to those who chose more disadvantageous risky choices. However, the magnitude of the relationship between risk-taking and noncompliant behavior was greater in the disadvantageous contexts than the equal contexts”. We have also made updates to the Abstract. 

Regarding the use of the terms ‘risk-neutral’, ‘risk-lover’, we understand that these terms are commonly used in the field of economics; however, in the field of psychology these terms are used less frequently. Psychological decision-making instead favors focusing on behavior (e.g., greater/less risk-taking; greater/less risk-aversion behavior) and avoiding the use of person-labels. The reason is simply that risk-taking behavior can be altered by many factors (emotions, stress, valuation differences, etc.) and a ‘risk-neutral’ person may be more risk-averse or more risk-loving in different states or situations. We agree that those that make a high proportion of disadvantageous risky choices are a minority of the population. We have now added a table (Table 2) to display the descriptive statistics for all the independent (including the proportion of risky choice) and dependent variables. Finally, in the Limitations section of the Discussion, we describe the limitations in generalizability by stating: “Given that the risk-taking findings were specific to equal and disadvantageous expected value contexts, it should be noted that some participants chose the safe option on all trials. Therefore, the relationship between risk-taking and compliance behavior may be driven by a subsample of the sample, which may limit the generalizability of the findings”. 

These revisions serve to explain the importance of examining both sensitivity to expected values and risk-taking as well as to better explain the risky choice task in the context of the broader psychology literature.

3) Another related point concerns the construction of the risk aversion variable. Dividing lotteries into these three levels and measuring the percentage of risky choices within each level seems a rather coarse approach. First, within each level, different lotteries will have different expected payoff values and thus different degrees of “advantageousness” and “disadvantageousness”. The level of risk is fixed in equal risky choice, but here (presumably), the size of the pie was manipulated. Hence, considerable information seems to have been ignored when constructing the indexes. The approach that I would advice is instead different. Drawing on contributions in experimental economics, a single parameter for individual risk aversion may be estimated, on the basis of choices throughout the three levels of risk. An approach models utility as “Constant Relative Risk Aversion”, and the curvature of the utility function (which is given by one parameter) is a synthetic indicator of an individual’s risk aversion (see Harrison & Rutström, 2008, Wakker, 2008). More sophisticated calibrations are also possible, including in particular the estimation of a loss aversion parameter (Abdellaoui et al., 2008). Incidentally, it is not clear from the text whether lotteries including losses were administered, but this seems to be the case from the examples reported in the questionnaire registred at OSF. Other approaches would be possible, but the current approach is unsatisfactory as it stands at the moment, in my view.

We realize that the task information was not clear in the original manuscript, and we thank the reviewer for noting this. We have now clarified in the Methods that the Risky Choice Task utilized in our task was similar to the Cups Task, which divides risky gambles into advantageous, disadvantageous, and equal expected value scenarios, and the study’s analytic approach mirrors the approach used with the Cups Task. We have now clarified this analytically approach in the manuscript by stating: “Following previous research using the Cups Task and similar risky choice paradigms (Brevers et al., 2012; Byrne & Ghaiumy Anaraky, 2020; Galván & McGlennen, 2012; Jasper et al., 2013; Levin et al., 2007; Weller et al., 2007; 2009; 2010; 2011; Yao et al., 2015), the proportion of risky gambles for each gamble type (risky advantageous, risky equal, and risky disadvantageous) was computed”. 

Additionally, we state that “the risky choice task involved 36 non-incentivized, hypothetical gain-framed decisions” in the manuscript to emphasize that there were not lotteries including losses administered. The full list of all 36 risky choices is now shown in Table S2 in the supplementary material. 

We emphasize that the purpose of this task and study is not to construct a risk aversion variable but rather to characterize the relationship between risky choices and mask-wearing/social distancing. We understand that the strength of constructing a risk aversion variable is to have a streamlined, valuable metric, and we have used computational modeling to create such parameters to characterize reinforcement learning behavior in other studies in our lab. However, these modeling parameters are typically supplemental to choices and/or decision-making performance, which is the focus of this study. Showing the differences in risky choice in disadvantageous vs. advantageous scenarios provides important information—it demonstrates the context in which risky decision-making is associated with mask-wearing and social distancing. By replacing the current risky choice variables with a single risk aversion metric, we would lose this information. 

4) The authors use linear regressions, but given the discrete nature of the dependent variable, an ordered logit model, or interval regression, would have been more appropriate. I am also not clear why authors use repeated measures for the risky hypothetical choices, done over the three levels (advantageous, equal, disadvantageous) – which, incidentally, was not part of the pre-analysis plan. This seems to arbitrarily inflate the power of the risky decision variable, and does not appear to be grounded in theory. Tables with the regression results should be reported, either in the main text or the Appendix.

All of the dependent variables are continuous variables, rather than discrete, which is why linear regressions were conducted. The addition of Table 2, which shows the range, mean, and standard deviation of the dependent variables should clarify this for readers.

While the OSF does not list repeated measures regression under the Analysis Plan, we do state in the pre-registration that Risk-taking score A (advantageous gambles), Risk-taking score B (disadvantageous gambles, and Risk-taking score C (ambiguous gambles) will be used as predictors in the Sampling Plan and Measured Variables sections. 

The repeated measures approach was used because risky-choice type (advantageous vs. disadvantageous vs. equal) is essentially a within-subjects moderator of risky choice. This approach is often used with the Cups Task and other risky decision-making tasks. To explain this approach, we state that “consistent with approaches used in similar risky choice tasks (Byrne & Ghaiumy Anaraky, 2020; Byrne et al., 2020; Levin et al., 2007; Madan et al., 2015; Weller et al., 2011), the within-subjects variable was defined as Risky Choice Level depending on whether the expected value for the risky option was higher, lower, or the same as the expected value of the safe option. The levels were operationalized as advantageous risky choice, disadvantageous risky choice, and equal risky choice” (p. 21). 

The tables with the regression results are now reported in the Supplementary Material. 

5) At page 16 the authors state: “equal gambles (N=12) in which the expected value for the risky and sure options were identical or nearly identical […]”. It appears arbitrary to classify lotteries whose expected value is the same or “nearly the same” as the certain option as belonging to the same category. In this sense, having 12 different lotteries that are “equal” seem to be rather excessive. I understand this may be customary in the strand of literature the authors are following. If so, this aspect should be clarified.

 We have added the following information to the Method section to clarify the risky choice scenarios: “While the Cups Task involves 54 gain and loss trials of varying expected value levels (disadvantageous, advantageous, and equal), the present task used a modified gains-only task because the study predictions were localized to risk behavior and not loss aversion. 

6) Page 16-17: “The Appendix shows the full list of questions”. I could only find three questions in the OSF website.

The Appendix on the OSF website originally included some examples of the type of questions that risky choice tasks often involve, rather than the comprehensive list. We have added the full list of questions to the Supplementary Material (Table S2), and we have updated the OSF website with the full list of Risky Choice Task questions. 

7) An obvious concern is that all the variables are self-reported. In particular, it was not clear to me that the choice of risky lotteries had not been monetarily incentivized. Only in the pre-registration of hypothesis I could eventually find this information. To the very least, the authors should discuss the implications of hypothetical Vs. monetarily incentivized questions (e.g. Beattie & Loomes, 1997; Brañas-Garza et al., 2020; Carlsson & Martinsson, 2001; Donkers et al. 2001).

We appreciate the reviewer noting this and providing helpful citations. We now state in the description of the Risky Choice Task that the task “involved 36 non-incentivized, hypothetical gain-framed decisions”. Additionally, we have added information on prior research that has compared hypothetical vs. real reward incentives to the Temporal Discounting and Risky Choice Task sub-sections of the Methods. These additions serve to clarify why hypothetical temporal and risky choice experimental measures are valid proxies of incentivized ones. Furthermore, we now acknowledge the Limitations of self-report variables in the Discussion by stating, “as with many studies that utilize self-report measures, social desirability biases may have influenced participants’ risky choices or disclosure of mask-wearing and social distancing behavior.” 

8) I appreciate that authors followed the good practice of pre-registering their hypotheses. It is clear that the analysis reported in the paper generally followed the pre-analysis plan. Nevertheless, there appear to be some deviations, and these should be flagged out in the paper. In particular, (a) the second hypothesis (Greater stress-related uncertainty due to COVID-19 will be associated with decreased risk-taking) has not been analyzed in the paper. (b) Dependent variable (3), “Willingness to return to work (continuous scale)” has not been analyzed, while “Perceived Risk” has been. I am very much against the pre-registration acting as a straitjacket on what authors should report in the paper. But transparency requires to inform the reader as to why some modifications of the pre-analysis plan were undertaken, what is post-hoc rationalization of results, and what is post-diction rather than pre-diction.

We acknowledge that we did not have a direct 1:1 correspondence between the OSF pre-registration, which occurred pre-IRB approval, and our final study design, and we neglected to document those differences in the manuscript. We have added the results showing the nonsignificant association between stress-related uncertainty due to COVID-19, risky choice, mask-wearing, and social distancing to the manuscript. Thus, the data do not support this hypothesis. The Perceived Risk variable is part of the original pre-registration (under Sampling Plan and Measured Variables). 

We collected the data for the willingness to return to work variable, but post-hoc realized that this variable was very different from the others (in terms of topic and analysis) and simply didn’t fit cohesively with the study focus. We have not analyzed any data for this variable, but the data will be made available on the OSF so that interested readers can pursue it if they wish. 

Finally, and most importantly, we have added documentation of changes from the OSF pre-registration in the Supplementary Materials. We refer to this in the Method section (p.12) by stating that “some deviations from the OSF pre-registration to the final study were made, such as the addition of the social distancing variables. Documentation of these differences are described in the Supplementary Material for full transparency”. 

9) The authors may benefit from including in their analysis the report issued by “The Covid States Project” (https://covidstates.org/), led by Northeastern University. In particular, the report on COVID19-associated behavior signals that wearing face masks is the only behavior that, among those considered, has been on the rise, while others, like social distancing, have been declining since measuring began (at the end of April). See: https://covidstates.org/
https://kateto.net/covid19/COVID19%20CONSORTIUM%20REPORT%2026%20BEHAVIOR%20NOV%202020.

We appreciate the reviewer’s suggestion to incorporate the findings of this report into the manuscript. To do so, we have first, added a sentence to the Method section stating that the social distancing metric used in the COVID States Project Report “is similar to the COVID States Project’s Relative Social Distancing Index (Lazer et al., 2020)” (p. 14). 

Since we increased the sample size and collected the second wave of data 2.5 months after the initial data was collected, we were able to examine whether mask-wearing and social distancing were different between the two time points (early September and end of December). We have added these results to the Descriptives section, and we include time point (September vs. December as a covariate in the analyses). The results showed that participants in the December data collection wave reported greater mask-wearing and social distancing than those in the September data wave. We discuss these findings in the Discussion section (p. 32) by stating, “Outside of political affiliation, the results indicated that those that participated in the study in December reported increased mask-wearing and social distancing than those that participated in September. This result is consistent with research from the COVID States Project (Lazar, 2020)”. 

Minor points:

10) The authors give the impression to associate “rationality” with decisions taken according to Expected Utility Theory (page 5). This is incorrect, and it does not seem to be necessary, also in the light of the authors’ following discussion. I do not see the need to incorporate the discussion of prospect theory in the introduction, as this theory is not used later on.

We have removed all references to Expected Utility Theory, Prospect Theory, and rationality/irrationality (in reference to decision-making under risk) from the Introduction.

11) It is really odd that a small portion of the sample (N=20) is made up of Clemson university students.

We have removed the Clemson University student sample from the analyses and recruited an additional 200 participants upon Reviewer 1’s suggestion. The final sample is now comprised of 404 MTurk-only participants. 

12) I find the paper too wordy in the introduction and discussion. There is no need to motivate the scope of the paper multiple times, or to expand theoretical review much beyond what is actually used in the paper. The policy implication discussed in the discussion to send out “positive messages” of what people may do, rather than negative messages of what people can’t do, does not seem to be supported by evidence produced in this paper.

In the Introduction, we have removed duplicate sentences regarding the scope and purpose of the study, substantially shortened the section on Decision-Making under Risk by removing the comparisons of expected utility and prospect theories, and shortened the Current Study and Hypotheses sub-section.

In the Discussion, we have removed the implications for messaging and shortened the overall implications. 

13) There are several inaccuracies: Page 5: “Thus, the probabilities of COVID-19 infection rates are known”. Strictly speaking, this is not true, because the actual cases are arguably much higher, and unknown, than reported cases. The modalities of infection are still not fully known.

We have removed this sentence from the manuscript. 

14) The author states “It is unclear how perceived risk will influence actual COVID-19 preventative behavior” (Page 6) and “It is unclear how individual differences in temporal discounting relate to COVID-19 preventative behaviors”. (Page 7). But at page 10 they say they hypothesize that higher perceived risk and higher temporal discounting are linked with lower preventative behavior (as should reasonably be expected).

In these sentences, we were conveying that the effect of perceived risk and temporal discounting on COVID-19 preventative behavior is unclear because they had not been empirically tested before this study. We predict that lower perceived risk and higher temporal discounting may be linked with lower preventative behavior, but this effect was not known before we obtained the results. 

However, to clarify this, we have changed “It is unclear how perceived risk will influence actual COVID-19 preventative behavior” to “It is expected that individuals with first-hand COVID-19 experience will have higher risk perception and that higher risk perception will enhance COVID-19 preventative behavior.” We have also removed the sentence stating “It is unclear how individual differences in temporal discounting relate to COVID-19 preventative behaviors” to make the paper less wordy as we state the hypotheses in the Current Study and Hypotheses sub-section. 

15) Page 9: “Because there is currently no vaccine available”: This should obviously now be updated.

We have replaced this phrase with the following statement: “At the time this research was conducted, COVID-19 vaccines were not available to the general population…”.

16) In the questionnaire text reported in the OSF website, the last question reads: “There’s a 75% chance that you will lose $100, but a $25 chance that you will not lose any money.” I hope the typo was amended when presented to participants.

This question was an example of the type of questions included in typical Risky Choice Tasks. We only included gain-framed, rather than loss-framed questions in the study procedure. We have updated this information on the OSF website, and the Supplementary Material now includes the full list of Risky Choice Task questions used in the study.

17) Page 19: Truncated sentence: “To examine whether the optimism bias, a paired samples t-test was conducted”.

We have corrected this sentence by removing the word ‘whether’. 

18) Page 11: The discussion of the quality of M-Turk samples is presented before saying that the sample was from M-Turk.

We appreciate the reviewer noting this, and we have now corrected this in the manuscript. 

19) Page 2: the authors claim that the sample is “representative”, but with N=225 this can never be the case.

We have removed the reference to the sample being representative.

References

Abdellaoui, M., Bleichrodt, H., & l’Haridon, O. (2008). A tractable method to measure utility and loss aversion under prospect theory. Journal of Risk and uncertainty, 36(3), 245.

Beattie, J., & Loomes, G. (1997). The impact of incentives upon risky choice experiments. Journal of Risk and Uncertainty, 14(2), 155-168.

Brañas-Garza, P., Jorrat, D., Espín, A. M., & Sanchez, A. (2020). Paid and hypothetical time preferences are the same: Lab, field and online evidence. arXiv preprint arXiv:2010.09262.

Carlsson, F., & Martinsson, P. (2001). Do hypothetical and actual marginal willingness to pay differ in choice experiments?: Application to the valuation of the environment. Journal of Environmental Economics and Management, 41(2), 179-192.

Cheng, Vincent CC, Shuk-Ching Wong, Vivien WM Chuang, Simon YC So, Jonathan HK Chen, Siddharth Sridhar, Kelvin KW To et al. "The role of community-wide wearing of face mask for control of coronavirus disease 2019 (COVID-19) epidemic due to SARS-CoV-2." Journal of Infection (2020).

Donkers, B., Melenberg, B., & Van Soest, A. (2001). Estimating risk attitudes using lotteries: A large sample approach. Journal of Risk and uncertainty, 22(2), 165-195.

Harrison, G. W., & Rutström, E. E. (2008). Risk aversion in the laboratory. Research in experimental economics, 12(8), 41-196.

Wakker, P. P. (2008). Explaining the characteristics of the power (CRRA) utility family. Health economics, 17(12), 1329-1344.

---

## [Decision Letter · Decision Letter 1]

23 Feb 2021

PONE-D-20-32678R1

Risk-Taking Unmasked: Using Risky Choice and Temporal Discounting to Explain COVID-19 Preventative Behaviors

PLOS ONE

Dear Dr. Byrne,

Thank you for submitting your manuscript to PLOS ONE. After careful consideration, we feel that it has merit but does not fully meet PLOS ONE’s publication criteria as it currently stands. Therefore, we invite you to submit a revised version of the manuscript that addresses the points raised during the review process.

As you will see referee #2 is still asking a number of serious modifications in the statistical analysis. Please do it carefully since I will send back the manuscript to him.

We look forward to receiving your revised manuscript.

Kind regards,

Pablo Brañas-Garza, PhD Economics

Academic Editor

PLOS ONE

Reviewers' comments:

Reviewer's Responses to Questions

**Comments to the Author**

1. If the authors have adequately addressed your comments raised in a previous round of review and you feel that this manuscript is now acceptable for publication, you may indicate that here to bypass the “Comments to the Author” section, enter your conflict of interest statement in the “Confidential to Editor” section, and submit your "Accept" recommendation.

Reviewer #1: All comments have been addressed

Reviewer #2: (No Response)

2. Is the manuscript technically sound, and do the data support the conclusions?

Reviewer #1: Yes

Reviewer #2: Partly

3. Has the statistical analysis been performed appropriately and rigorously? 

Reviewer #1: Yes

Reviewer #2: I Don't Know

4. Have the authors made all data underlying the findings in their manuscript fully available?

Reviewer #1: Yes

Reviewer #2: No

5. Is the manuscript presented in an intelligible fashion and written in standard English?

Reviewer #1: Yes

Reviewer #2: Yes

6. Review Comments to the Author

Reviewer #1: Thanks to the authors for adressing all the comments. The paper impoved substantially. Congratulations.

Reviewer #2: See attached report. See attached report. See attached report. See attached report. See attached report.

7. PLOS authors have the option to publish the peer review history of their article (what does this mean?). If published, this will include your full peer review and any attached files.

Reviewer #1: **Yes: **Diego Andrés Jorrat

Reviewer #2: No

---

## [Author Response · Author response to Decision Letter 1]

29 Mar 2021

The present version of the paper is improved and addresses most of the issues that the other reviewer and I had raised. I also accept that there may be differences in methods and approaches between psychology and economics, which should be toned down in an interdisciplinary journal like PLOS-1. Nonetheless, the authors do choose economic theory to justify their methods, and should then make sure that their approach is correct. Most importantly, I am still unclear on the nature of the econometric model the authors use. I would then recommend the authors to address the following issues:

1) The authors inserted the following statement to explain their approach concerning the measurement of risk aversion (pp. 4-5):

Choosing risky options with higher expected utility is economically advantageous, while choosing risky options that have a lower expected utility than a safe option is maladaptive and can often lead to sub-optimal decision-making (Weller, Levin, & Bechara, 2010; von Neumann & Morgenstern, 1944). I find this statement confusing and factually wrong. I suspect that the authors confuse expected utility with the expected value of a gamble (i.e. a risky choice). Otherwise, taking this sentence literally would entail that individuals would choose “risky options that have a lower expected utility than a safe option”. Since utility is unobservable, the principle of revealed preferences by von Neumann and Morgestern (cited) – after the rationalization by Savage (1954) -, states that, precisely for this reason, we can deduct the expected utility function from actual choices that people make. Observing individuals choosing lotteries having lower expected utility than a safe option is then by construction impossible within the theory. This is why I guess that the authors meant “expected value” rather than expected utility, as also suggested by the language they use in other parts of the paper (e.g. figures’ captions). Even so, their statement is factually incorrect. While saying that lotteries with higher expected value are economically advantageous is a platitude, saying that

“choosing risky options having a lower expected *value* than a safe option is maladaptive and can often lead to sub-optimal decision-making” goes against cross-country experimental evidence finding that economically richer countries are characterized, on average, by lower, rather than higher, risk

tolerance (Falk et al., 2018; Bouchouicha, R., & Vieider, 2019). Admittedly, this finding only holds after controlling for other psychological traits (see Table 9 in Falk et al., 2018), while an opposite

relation between risk tolerance and income holds within country (l'Haridon & Vieider, 2019).

Nevertheless, this evidence is enough to make the authors’ statement basically unfounded. Moreover, it is perfectly acceptable, I would dare say from a psychological point of view, that individuals prefer a safe option to a gamble, be it advantageous or disadvantageous, because it is perfectly acceptable that for most individuals well-being (or utility) is decreased by risk. This should not be seen as either maladaptive or sub-optimal, as it is in fact a rather widespread characteristic of individual preferences.

In sum, I find the authors’ justification of their multi-level approach used in their econometric analysis to be flawed theoretically and empirically. I would suggest the authors to drop any reference to expected utility theory. I would instead ask the authors to provide a different theoretical justification of the reason why we should consider choices in the disadvantageous, equal, or advantageous domain, as distinct from one another, possibly relying on other studies doing so.

To correct this issue in the Introduction, we have reframed this paragraph by removing all references to expected utility. We conceptual adaptive or optimal decision-making in terms of reward maximation, which depends on sensitivity to expected values (not utility, in accordance with what the comment above describes), rather than utility or individual preference. This conceptualization is in line with the 7,000+ studies that have relied on tasks like the Cups Task and Iowa Gambling Task (Bechara et al., 1994) and the Expectancy Valence model of behavior on these tasks (Yechiam et al., 2005; Worthy, Pang, & Byrne, 2013), which consistently shows that greater sensitivity to expected values is associated with greater reward maximization. Indeed, the Cups Task was developed in collaboration with Bechara, who developed the IGT. 

We are a bit unclear on the comparisons between the references cited on the relationship between income and risk tolerance and the assumption that choosing higher expected values in classic risky choice decision-making paradigms leads to adaptive decision-making (i.e., decision-making that maximizes reward). We have statistically controlled for income by including income level as a covariate in the regressions. Moreover, behavior that leads to reward maximization is quite different from the demographic variable of income, which is largely a product of one’s life circumstances, privilege (i.e., white privilege), and inequality in access to resources. 

We have updated the paragraph to instead rely on the same framework as the one used in the seminal Cups Task paper: sensitivity to expected values. We replace the sentence described above with the following (p. 5): “Choosing options with higher expected values reflects increased sensitivity to differences in expected value between choice options [7, 8]. As evidenced by performance on decision-making paradigms such as the Iowa Gambling Task [9, 10] and the Cups Task [7,8], this increased sensitivity can lead to reward maximization”. In other words, choosing the risky choice in advantageous expected value contexts but not disadvantageous expected value contexts reflects increased reward sensitivity and is likely to lead to reward maximation. Using this task allows us to assess both risk-taking behavior (overall performance on the Cups Task) and sensitivity to expected values (varying behavior in different expected value contexts). This explanation was already expressed in the Discussion (p. 30, top), but we failed to make this information clear in the Introduction originally. However, given the comments below, we’ve toned down the emphasis on risky choice in these varying contexts. 

2) Even if the reader can now have a clear view of how the measurement of risk-taking behaviour

was carried out, thanks to Table S2 in the Appendix, I still find their econometric model unclear. 

Have the authors taken the three following measures for each individual: [proportion of risky

choices taken in the disadvantageous domain; proportion of risky choices in the “equal” domain;

proportion of risky choices in the advantageous domain]? This is what appears from the descriptive statistics. If this is the case, this should be stated clearly. But then, what is the purpose of including what appear to be interaction terms between the “level” of the variables, as defined above, and the proportion of risky choices? This is what appears in Table S3 in the Appendix, while the caption seems to indicate the lack of inclusion of interaction terms:

Equal EV Gamble X Equal Risky Choices indicates proportion of equal EV risky choices and Disadv. EV

Gamble X Disadv. Risky Choices refers to proportion of disadvantageous EV risky choices.

 But then, why do the authors include these five variables in the model?

Risky Choice EV Type

Equal EV Gamble

Disadvantageous EV Gamble

Equal EV Gamble X Equal Risky Choices

Disadv. EV Gamble X Disadv. Risky Choices

By reading that the authors used a repeated measure model, my understanding was that the authors

organized their data in panel format, with the individual as the cross-sectional unit and the three

decisions at the three levels as the “longitudinal” variable. But having these interaction terms in the

model (assuming they are interaction terms) makes me think that the authors used instead a pooled

model? And in any case, what is the point of having an interaction term between the different levels?

Are we at all interested in knowing that participants made riskier decisions in the advantageous

domain? This seems obvious, and I have not seen this result being reported. In conclusion, I think it is

necessary that the authors provide more details of their econometric approach, written down with

equations to avoid any misunderstanding. I would also appreciate if the authors could make available their data and codes they used in their analysis. This is requested in any case by publication in PLOS1. The authors should also report in Table S3 the total number of observations, and the observations by cluster (three, if my interpretation is correct). Since observations are clustered at the individual level, the best practice is to use heteroschedasticity-robust standard errors clustered at this level.

We acknowledge that the presentation of the regressions was unclear, and we have now added a “Data Analysis” subsection to the paper to clarify the statistical approach for the regressions. Originally, the proportion of risky choices was included as a main effect, Expected Value Level (dummy coded) was also a main effect, and then the Proportion of Risky Choices X Expected Value Level interaction term was included, which allowed for assessing whether risky choices in the advantageous, disadvantageous, or equal gamble domains were predictive of each dependent variable. There were 5 terms included in the tables because MPlus and R (the statistical analysis tools we use), have options to combine the main effects and simple contrasts to identify the locus of the interaction into one table, which can be more informative. The rationale for including the different levels was to show the contexts in which participants made risky choices—essentially does all risky choice or just risky choices that leads to lower expected values (i.e., disadvantageous) affect mask-wearing and social distancing? To answer this question, it is important methodologically to include advantageous risky choices as a comparison group. 

However, based on these comments, we can see where this approach may ‘muddy the waters’ and make the analyses unnecessarily complex without adding substantial knowledge depth. As such, we have now removed the analysis by EV level from the regression models. Now, average proportion of risky choices (averaged across advantageous, disadvantageous, and equal contexts) alone is included in the model, and multiple regressions are used. We have also added the regression equation. This information is now explained in the new Data Analysis sub-section (p. 18). 

Furthermore, in the description of the risky choice task, we now state “the average proportion of risky gambles across all gambling types was computed for regression analyses and used as the primary analysis variable for this task. Additionally, following previous research using the Cups Task and similar risky choice paradigms [8, 63—70], the proportion of risky choices for each gamble type (risky advantageous, risky equal, and risky disadvantageous) was computed and used in follow-up analyses; this provides further information about whether sensitivity to expected values, as reflected by differential risk-taking in advantageous compared to disadvantageous decision contexts, influences the outcome variables” (p. 16, bottom & p. 17, top). 

The data is available; we included the link in the PLoS One submission questions but didn’t realize this information was not sent to reviewers; we apologize for our misunderstanding. The link to the data is here: 

https://osf.io/xy6aj/?view_only=1360495284da453baac8df96c7a732d1

Upon manuscript acceptance, we will remove the ‘view-only’ link and the link will be https://osf.io/xy6aj/. This link has now been added to the manuscript. We are hesitant to make the data available outside of a view-only link before manuscript acceptance because we have had our data stolen by an outside research group in the past. 

We have now added the number of observations/participants and heteroscedasticity-robust standard errors (Huber-White standard errors) to Tables S3 – S7 in addition to the R2-value and p-value for the omnibus test.

In terms of addressing the specific statistical concerns, we note that our analytical expertise is rooted in training from experimental psychology and computer science. Therefore, we are unfamiliar with some of the terminology referred to (e.g., econometric model, cross-sectional units, pooled models, etc.), as this lies outside of our field. We mention this simply to acknowledge that we may use different terms to refer to similar concepts/approaches that exist in other fields, such as economics.

3) As could have been expected, only a minority of people chose risky options in the

disadvantageous and equal domain, while more chose the risky option in the advantageous

domain. The authors discuss this aspect in the discussion section:

“Given that the risk-taking findings were specific to equal and disadvantageous expected value contexts, it should be noted that some participants chose the safe option on all trials. Therefore, the relationship between risk-taking and compliance behavior may be driven by a subsample of the sample, which may limit the generalizability of the findings”. 

This statement sounds unnecessarily reticent. Rather than saying that the results are driven by

“some participants”, the authors should quantify how many participants chose the safe option in all cases. I would expect a more extensive discussion of the fact that the result is driven by only a

minority of participants. This is not at all uncommon in statistical analysis. Moreover, for social

phenomena like mask-wearing, the behavior of a minority individuals, maybe as low as 10% of the

population, may be enough to tip the social equilibrium from one of general compliance to one of

general non-compliance. Therefore, I do not think that even if the result was driven by a minority of

participants would detract from the general interest of the paper.

We have now added the percentage of participants who chose the safe option in all cases to the Results section (p. 20, bottom). In particular, we state: “In terms of risky decision-making, 17.08% of participants chose the safe option on all trials. By gambling context, 19.55% of participants chose the safe option in all advantageous expected value contexts, 46.78% chose the safe option in all equal expected value contexts, 57.43% chose the safe option in all disadvantageous contexts.” 

We have updated the Discussion to include a more extensive discussion of this finding (p.30, middle). Specifically, we have added the following text: “It should be noted that the average percentage of risky choices across all participants in equal and disadvantageous contexts was low—less than 20%. Moreover, over half (57%) of participants chose the safe option on all disadvantageous trials, and 47% chose the safe option on all equal expected value trials. The relationship between risk-taking and compliance behavior appears to be driven by a minority of participants, which may limit the generalizability of the findings. Nevertheless, the behavior of even a minority of individuals can have substantial impacts during a pandemic. Recent estimates indicate that approximately 10% of infected individuals account for 80% of COVID-19 transmission [75]. Though a minority of individuals engage in equal and disadvantageous risky decision-making, the corresponding decrease in compliance with mask-wearing and social distancing observed in this study could have wide-ranging consequences for spreading COVID-19.

4) Even if the above clarifications are necessary in my opinion, I still struggle to see the need for breaking down the observations by domain (advantageous, equal, disadvantageous gambles).

What is this regression is telling us? Is it really telling us that people who are more risk tolerant,

as measured in the “cups task”, wear the mask less frequently? This would seem the type of

analysis to address the research question of the paper. For example, suppose that Participant P1 chose all risky options in the advantageous domain, and all safe options in the equal and disadvantageous domain. Suppose that Participant P2 chose all safe options in the advantageous domain and in the equal domain, while choosing a quarter of risky choices in the disadvantageous domain. The econometric model would predict that P1 should wear a mask in public, while P2 should not. But can we infer in any meaningful sense that this kind of choice captures that P2 is more risk tolerant than P1? Or is this finding relevant purely on methodological grounds, rather than to associate individual psychological traits to individual real-life behavior?

 To me it would seem more natural to consider one measure of risk tolerance for each individual

defined over all the possible set of lotteries. Even if more sophisticated measures would be

possible, an obvious candidate would be to use the proportion of risky choices over the whole

set of 36 choices. This measure should be used as the main predictor in a regression analogous

to the one reported in Table S3. Looking at the correlations in Table 4, it would appear that this

variable would turn out as a significant predictor of mask-wearing. The authors could

subsequently break down the analyses by domain, as they presently do.

We appreciate the feedback on this point. To briefly explain our original rationale, the original regression was intended to gauge whether individuals’ risky decisions on the Cups Task (in addition to other factors) relates to their decisions regarding mask-wearing and social distancing. The risk level (advantageous, disadvantageous, and equal expected value) for this task is typically included as a modifier—simply to provide more specific information about the contexts in which people make or do not make risky choices (disadvantageous vs. equal, etc.). Those that make riskier choices in advantageous contexts more than disadvantageous contexts are considered more sensitivity to changes in expected value [7, 8]. The task and analytical approach were intended to capture this context variability because risk-taking is not a static ‘trait’ but varies within a single individual based on domain, context, and time (Weber et al., 2002), while psychological traits are defined as consistent over time and context. We realize that this information was not conveyed clearly in the previous revision. 

Based on the feedback, we have now revised the regressions for each outcome measure to include a single predictor of risky decision-making behavior, which is defined as the proportion of risky choices over the whole set of choices as suggested. After reporting the results of the primary regressions, we report the results of separate follow-up tests that break down the analyses by gambling domain. The results showed that both risky decision-making behavior and temporal discounting predicted decreased mask-wearing and social distancing behavior. There were changes in the significance level for some of the covariates, and this information has been updated throughout the results and discussion. We have also now included some brief additional information to better explain why the different contexts (advantageous, disadvantageous, and equal) were examined as part of the study (p. 5, top). The corresponding tables have also now been updated. Overall, this updated approach led to more straightforward results that we think may be clearer and more useful for a broad audience like the PLoS One readership to interpret. 

Weber, E. U., Blais, A. R., & Betz, N. E. (2002). A domain‐specific risk‐attitude scale: Measuring risk perceptions and risk behaviors. Journal of behavioral decision making, 15(4), 263-290.

5) I note that the authors went at great length in addressing the comment that pro-sociality was a

possible confound in the theoretical model. I agree with the authors that giving too much

attention to this aspect would detract from the main focus of the paper. Nonetheless, since both

the other reviewer and I had the same concern, it is quite likely that other readers would as

well. I would then recommend the authors to report their analysis concerning pro-sociality in

the Appendix and mention the result in the main paper. I would also stress that it is obvious that

the list of possible explanatory variables is potentially infinite, and nobody in their right mind

could ask to incorporate all of them in the analysis. Nonetheless, pro-sociality is such an obvious

possible motivator in the current context, that many would feel that not including could

jeopardize the interpretation of the results. 

We have now added the description of the prosociality measures to the Methods section (p. 17). We then elaborate on the methodological details and report the results in the Supplementary Material and mention the correlational results in the Results section of the main paper (p. 29) and in the Discussion (p. 34). The sentence stating that an exhaustive list of possible influences on COVID-19 compliance has also been removed.

---

## [Decision Letter · Decision Letter 2]

20 Apr 2021

Risk-Taking Unmasked: Using Risky Choice and Temporal Discounting to Explain COVID-19 Preventative Behaviors

PONE-D-20-32678R2

Dear Dr. Byrne,

We’re pleased to inform you that your manuscript has been judged scientifically suitable for publication and will be formally accepted for publication once it meets all outstanding technical requirements.

Kind regards,

Pablo Brañas-Garza, PhD Economics

Academic Editor

PLOS ONE

Additional Editor Comments (optional):

Reviewers' comments:

Reviewer's Responses to Questions

**Comments to the Author**

1. If the authors have adequately addressed your comments raised in a previous round of review and you feel that this manuscript is now acceptable for publication, you may indicate that here to bypass the “Comments to the Author” section, enter your conflict of interest statement in the “Confidential to Editor” section, and submit your "Accept" recommendation.

Reviewer #2: All comments have been addressed

2. Is the manuscript technically sound, and do the data support the conclusions?

Reviewer #2: Yes

3. Has the statistical analysis been performed appropriately and rigorously? 

Reviewer #2: Yes

4. Have the authors made all data underlying the findings in their manuscript fully available?

Reviewer #2: Yes

5. Is the manuscript presented in an intelligible fashion and written in standard English?

Reviewer #2: Yes

6. Review Comments to the Author

Reviewer #2: (No Response)

7. PLOS authors have the option to publish the peer review history of their article (what does this mean?). If published, this will include your full peer review and any attached files.

Reviewer #2: No

---

## [Editor Report · Acceptance letter]

3 May 2021

PONE-D-20-32678R2 

Risk-Taking Unmasked:Using Risky Choice and Temporal Discounting to Explain COVID-19 Preventative Behaviors 

Dear Dr. Byrne:

I'm pleased to inform you that your manuscript has been deemed suitable for publication in PLOS ONE. Congratulations! Your manuscript is now with our production department. 

Kind regards, 

on behalf of

Dr Pablo Brañas-Garza 

Academic Editor

PLOS ONE